# Combined IgE neutralization and *Bifidobacterium longum* supplementation reduces the allergic response in models of food allergy

Seong Beom An [1,2,10], Bo-Gie Yang [3,4,10] ✉, Gyeonghui Jang[3], Do-Yeon Kim[3], Jiyoung Kim[3], Sung-Man Oh[3], Nahyun Oh[3], Sanghee Lee[3], Ji-Yeong Moon[5], Jeong-Ah Kim[3], Ji-Hyun Kim[3], Yoo-Jeong Song[3], Hye-Won Hyun[3], Jisoo Kim[3], Kyungwha Lee[3], Dajeong Lee[4], Min-Jung Kwak[4], Byung Kwon Kim [4], Young-Kyu Park [4], Chun-Pyo Hong[3,6], Jung Hwan Kim[7], Hye Seong Lim[7], Min Sook Ryu[5], Hyun-Tak Jin[7], Seung-Woo Lee[1,2], Yoon-Seok Chang[8], Hae-Sim Park [5], Young Chul Sung [1,2] ✉ & Myoung Ho Jang [3,9] ✉

IgE is central to the development of allergic diseases, and its neutralization alleviates allergic symptoms. However, most of these antibodies are based on IgG1, which is associated with an increased risk of fragment crystallizable-mediated side effects. Moreover, omalizumab, an anti-IgE antibody approved for therapeutic use, has limited benefits for patients with high IgE levels. Here, we assess a fusion protein with extracellular domain of high affinity IgE receptor, FcεRIα, linked to a IgD/IgG4 hybrid Fc domain we term IgE_TRAP, to reduce the risk of IgG1 Fc-mediated side effects. IgE_TRAP shows enhanced IgE binding affinity compared to omalizumab. We also see an enhanced therapeutic effect of IgE_TRAP in food allergy models when combined with *Bifidobacterium longum*, which results in mast cell number and free IgE levels. The combination of IgE_TRAP and *B. longum* may therefore represent a potent treatment for allergic patients with high IgE levels.

IgE is an important target for the treatment of allergic diseases. An anti-IgE antibody, omalizumab, has been approved by the United States of America (USA) Food and Drug Administration (FDA) for the treatment of moderate to severe persistent allergic asthma, nasal polyps and chronic spontaneous urticaria (CSU)[1]. It even received Breakthrough Therapy designation to facilitate its development for the treatment of food allergy. However, omalizumab has been reported to have limited efficacy in lowering high serum IgE levels of atopic dermatitis patients[2]. Moreover, use of omalizumab has an associated risk of IgG-mediated passive anaphylaxis, which is thought to arise from omalizumab's IgG1 fragment crystallizable (Fc) domain binding to low-affinity IgG receptors (FcγRs)[3–8]. These concerns around the limited efficacy and safety

of omalizumab identify the need for a better anti-IgE agent for the treatment of IgE-mediated allergic diseases.

Recent reports have shown a close correlation between allergy and intestinal microbiota. Young children with atopy show decrease in intestinal bacteria belonging to *Bifidobacterium* and *Lactobacillus*, and infants with the resolution of food allergy to milk by the age of 8 years show increase in *Clostridium* bacteria in the intestine[9–12]. In murine models of food allergy, the administration of *Bifidobacterium longum* or *Clostridial* species alleviate allergic symptoms by either inducing mast cell apoptosis or regulatory T cell (Treg) activity[13–15]. Furthermore, atopic dermatitis patients who received oral treatment of *Bifidobacterium* and *Lactobacillus* species showed a reduction in the

---

allergic symptoms[16,17]. However, clinical trials involving probiotic therapies have shown only modest results, suggesting that combination approaches may be required.

In this work, we present IgE_TRAP molecule generated by linking the extracellular domain of human high-affinity IgE receptor, hFcεRI, to an IgD/IgG4 hybrid Fc domain. Since IgE does not easily dissociate from its native FcεRI receptor[18], IgE_TRAP has a conceptual advantage over the anti-IgE antibodies generated to date. The IgD/IgG4 hybrid Fc domain was constructed by linking part of the IgD Fc region, which is a long flexible hinge without the binding sites for the FcγRs and the complement component 1q (C1q), to the part of IgG4 with the binding site of neonatal Fc receptor (FcRn). The resultant Fc-fusion protein displayed a longer half-life from its interaction with FcRn, as well as low risk of antibody-dependent cellular cytotoxicity (ADCC), complement-dependent cytotoxicity (CDC), and IgG-mediated anaphylaxis from its lack of interaction with FcγRs. We further show that the therapeutic effect of IgE_TRAP is enhanced by the combined administration with anti-allergic probiotic *B. longum*[13], associated with the reduction in mast cell number and free IgE levels. Taken together, our study suggests the combined use of IgE_TRAP and anti-allergic probiotics as a safe and effective treatment for allergic patients with high IgE serum levels.

## Results

### IgE_TRAP is a fusion protein comprising a FcεRIα extracellular domain and an IgD/IgG4 hybrid Fc domain and is less likely to induce IgG1 Fc-mediated side effects

The high-affinity IgE receptor, FcεRI, features an α-chain, a β-chain, and two identical, disulfide-linked γ-chains. While FcεRIβ and FcεRIγ have no extracellular domains, FcεRIα has an extracellular immunoglobulin-related domain which is involved in IgE binding[19]. We linked the human FcεRIα extracellular domain to a human IgD/IgG4 hybrid Fc domain to produce IgE_TRAP (Fig. 1a, b), in order to block IgE binding to FcεRIα on effector cells. Given that the binding sites for FcγRs and C1q on IgG4 are located across the region proximal to the hinge and the upper region of the CH2 domain, the hybrid Fc chain was generated by integrating the hinge region and the upper region of the CH2 domain of IgD, which has no binding sites for FcγRs and C1q, to the lower region of CH2 domain and CH3 domain of IgG4 (Fig. 1a).

The IgD/IgG4 hybrid Fc lacks binding sites for FcγR and C1q, thereby reducing the likelihood of ADCC, CDC, and IgG-mediated anaphylaxis[4,20]. As a result, IgE_TRAP showed no significant binding to FcγRs and C1q (Fig. 1c). In contrast, omalizumab, an IgG1 antibody, bound to FcγRs and C1q (Fig. 1c). In support of this observation, NK cells expressing FcγRIIIA did not release any granzyme B when they were added to an IgE-coated plate with IgE_TRAP (Fig. 1d). In contrast, omalizumab and IgE_TRAP-IgG1M2, IgE_TRAP with a modified IgG1-Fc instead of the IgD/IgG4 hybrid Fc, strongly induced granzyme B secretion (Fig. 1d). In EpiScreen assays using CD8+-depleted peripheral blood mononuclear cells (PBMCs) from healthy human donors, IgE_TRAP showed low helper T cell response like omalizumab and trastuzumab that were used as negative clinical controls (<10% of donors responded) (Table 1, Supplementary Table 1, and Supplementary Fig. 1). IgE_TRAP did not show any T cell response in one experiment of the two independently conducted EpiScreen assays (Supplementary Table 1 and Supplementary Fig. 1). Meanwhile, exenatide and KLH, which were used as positive controls, showed strong T cell response in the two experiments (Table 1 and Supplementary Table 1). This result suggests that IgE_TRAP has a minimal risk of immunogenicity. Although the IgE_TRAP hybrid Fc domain retains FcRn binding site to enable a longer half-life[21,22], IgE_TRAP has a total of eight N-linked glycosylation sites which can enhance clearance of IgE_TRAP in vivo by binding to the glycan receptors. Thus, to extend the half-life of IgE_TRAP in vivo, α−2,6-sialyltransferase was co-transferred into Chinese hamster ovary (CHO) DG44 cells and the glycans on the IgE_TRAP was capped by sialylation. Due to the negatively-charged sialic

acids (SA), IgE_TRAP with high sialic acid contents produced a broad band between the isoelectric points (pI) values of 5.3 and 4.2 in isoelectric focusing (IEF) gel (Supplementary Fig. 2). Also, the molecular weight was observed to be ~150 kDa, even though the theoretical molecular weight of the homodimerized IgE_TRAP is ~97.6 kDa (Fig. 1e).

### IgE_TRAP is superior to omalizumab in blocking IgE binding to FcεRI and mast cell degranulation

A surface plasmon resonance (SPR) assay was performed to compare the binding affinities of IgE_TRAP and omalizumab to human IgE. The association rate constant ($K_a$) and dissociation rate constant ($K_d$) for IgE_TRAP were approximately 1.5- and 105-fold lower, respectively, than omalizumab (Fig. 2a). It has previously been reported that IgE dissociates slowly from FcεRIα[18], and likewise, IgE_TRAP showed slow dissociation of IgE, with a binding affinity for human IgE being 69-fold higher than omalizumab (Fig. 2a). To further examine the ability of IgE_TRAP and omalizumab to dissociate IgE-FcεRI complexes, Bio-Layer Interferometry (BLI) assay was performed by adding IgE_TRAP or omalizumab to human IgE pre-complexed with immobilized recombinant human FcεRIα. Omalizumab dissociated the IgE-FcεRI complexes as previously reported[23], however, IgE_TRAP elicited dissociation more rapidly (Fig. 2b).

A mast cell degranulation assay was also performed to assess the suppressive capacity of IgE_TRAP during IgE-mediated allergic reactions. The human mast cell line LAD-2 was used for this assay and the result showed that IgE_TRAP prevented mast cell degranulation more effectively than omalizumab (Fig. 2c). As omalizumab is non-crossreactive to mouse IgE, comparing its efficacy with IgE_TRAP was not feasible in the mouse models of allergy[24]. Since a correlation between suppression of free IgE level and clinical outcomes by a drug has been well reported[25], we took an indirect approach to compare the efficacy of IgE_TRAP and omalizumab by subcutaneously injecting them into cynomolgus monkeys with high serum IgE levels under basal conditions and then examining their free IgE levels in serum over time. Free IgE here refers to IgE that has the ability to bind to IgE receptors on effector cells without binding to IgE inhibitors and can cause allergic reaction. IgE_TRAP reduced serum levels of free IgE more effectively than omalizumab in a dose-dependent manner (Fig. 2d). Also, a single dose of IgE_TRAP at 10 mg kg⁻¹ (mpk) maintained its suppressive effect for 3 days, while 30 mpk and 60 mpk doses maintained the inhibitory effect for 7 and 10 days, respectively. In contrast, omalizumab failed to fully reduce the free IgE levels at its highest dose, even though the monkeys administered with omalizumab had lower basal serum IgE levels than monkeys administered with IgE_TRAP. Furthermore, inhibitory effect of omalizumab only lasted for 3 days even at the highest dose of 60 mpk (Fig. 2d). IgE_TRAP significantly reduced free IgE levels in 3 h, as opposed to 6 h for omalizumab at 60 mpk (Fig. 2d). In addition, IgE_TRAP more effectively inhibited levels of free IgE in the sera from CSU patients (Fig. 2e).

### The bioactivity of IgE_TRAP in vivo is affected by its sialic acid contents

To examine the effect of sialic acid on the efficacy of IgE_TRAP in vivo, first, IgE_TRAP with either high or low sialic acid contents (IgE_TRAP SA^high & IgE_TRAP SA^low) were separated using anion exchange chromatography. Due to the presence of negatively charged sialic acid molecules, IgE_TRAP SA^high showed a major band at lower pI values than IgE_TRAP SA^low in IEF gel (Supplementary Fig. 2). Since the most common type of sialic acid is N-acetylneuraminic acid (NANA), the sialic acid content is generally expressed by the NANA content[26]. IgE_TRAP SA^high and IgE_TRAP SA^low retained 20.9 moles and 11.4 moles of NANA per mole of protein, respectively (Supplementary Fig. 2). In an ovalbumin (OVA)-induced food allergy model causing acute diarrhea, the IgE_TRAP SA^high (19.7 mol mol⁻¹) significantly reduced the diarrhea occurrence and the serum levels of free IgE and Mast Cell Protease-1 (MCPT-1; a marker of

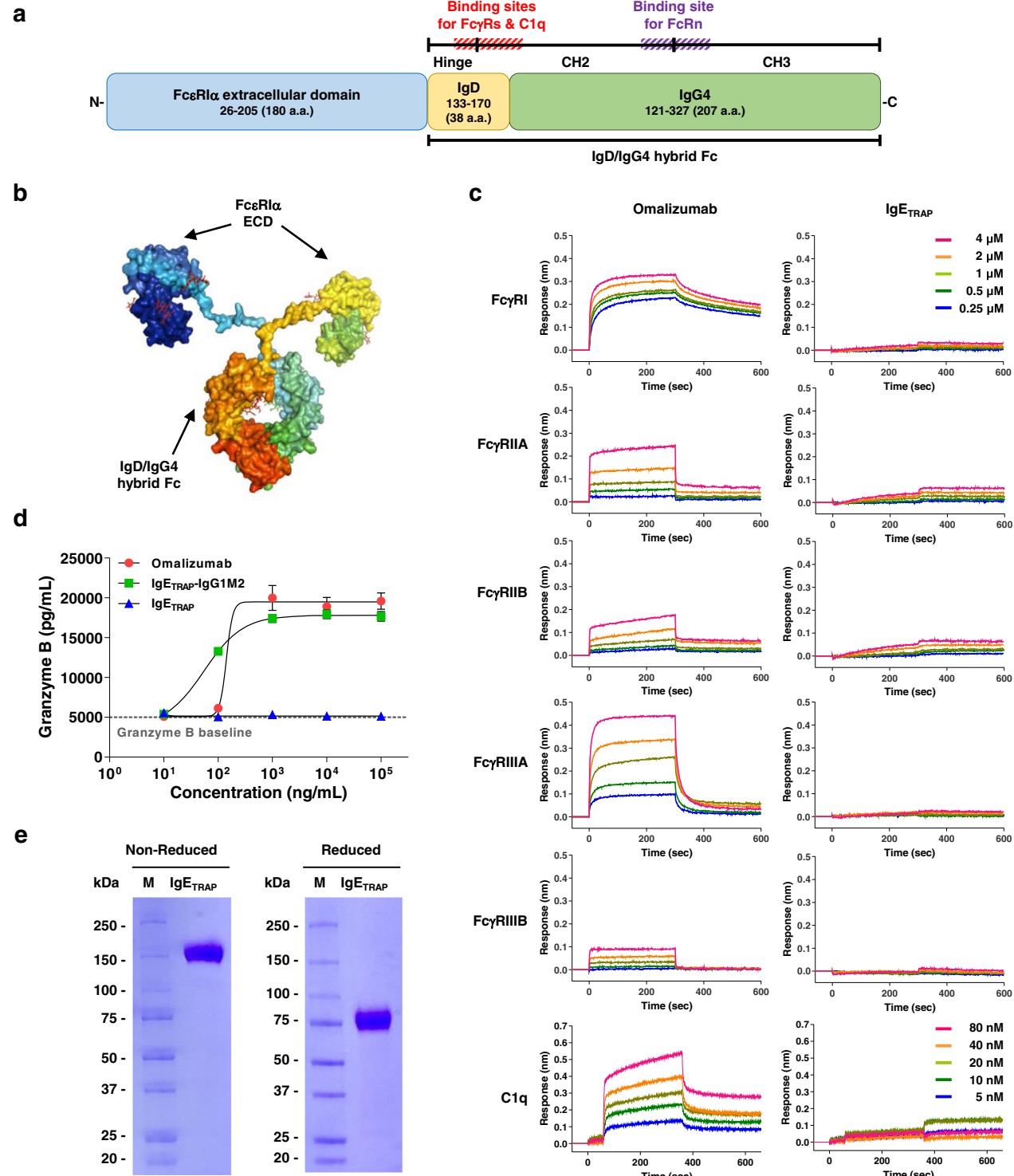

**Fig. 1 | Structural features of IgE_TRAP. a** Schematic diagram of the IgE_TRAP monomer. IgE_TRAP is composed of 425 amino acids from the extracellular domain of human FcεRIα (180 amino acids) fused to a human IgD/IgG4 hybrid Fc (245 amino acids). The IgD/IgG4 hybrid Fc region has an FcRn binding site (purple hatched line) but lacks FcγR and C1q binding sites (red hatched line). **b** 3-D structural model of the IgE_TRAP homodimer. The structure shows the FcεRIα extracellular domain, IgD hinge, and IgG4 Fc. **c** Interactions of IgE_TRAP and omalizumab with IgG receptors (FcγRI, FcγRIIA, FcγRIIB, FcγRIIIA, and FcγRIIIB) and C1q were assessed using a BLI assay. Representative data from two independent experiments. **d** Granzyme B levels released from NK cells by antibody-dependent cellular activation via FcγRIII binding. The figure is representative of three independent experiments. Data are presented as mean values ± SD. **e** SDS-PAGE analysis of IgE_TRAP was performed under non-reducing and reducing conditions. The figure is representative of three independent experiments. Source data are provided as a Source Data file.

**Table 1 | Episcreen assay**

| Sample | Mean SI* | SD | % Response |
|---|---|---|---|
| IgE$_{TRAP}$ | 2.78 | ± 0.63 | 9 |
| Omalizumab (Xolair) | 2.12 | ± 0.06 | 4 |
| Trastzumab (Herceptin) | 2.72 | ± 0.45 | 9 |
| Exenatide (Bydureon) | 2.99 | ± 1.62 | 43 |
| Keyhole limpet hemocyanin (KLH) | 11.78 | ± 9.26 | 96 |

N/A indicates no data available.

Stimulation Index (SI) = mean value of test wells (cpm)/baseline (cpm). The mean SI was calculated from the average of all positive donor responses observed during the entire time course (days 5–8).

mast cell degranulation), but the IgE$_{TRAP}$ SA$^{low}$ (7.7 mol mol$^{-1}$) did not (Fig. 3a, b, and c). In the mouse model of IgE-mediated passive systemic anaphylaxis, IgE$_{TRAP}$ with 10.3 mol mol$^{-1}$ of NANA suppressed the body temperature drop, but not to the extent seen with IgE$_{TRAP}$ with the higher NANA contents, 14.9 or 21.4 mol mol$^{-1}$ (Fig. 3d). Moreover, C$_{max}$, AUC$_{last}$ and AUC$_{0-\infty}$ of IgE$_{TRAP}$ increased with increasing sialic acid contents, indicating that the exposure of IgE$_{TRAP}$ in vivo increases according to the sialic acid contents (Fig. 3e and Table 2). These results show that sialylation reduces clearance of IgE$_{TRAP}$ and increases its bioactivity in vivo.

### *B. longum* enhances the therapeutic effect of IgE$_{TRAP}$ in food allergy models

To assess the effect of IgE$_{TRAP}$ on food allergy, an OVA-induced food allergy model that induces acute diarrhea in BALB/c mice was used (Fig. 4a and 5a). A single injection of IgE$_{TRAP}$ effectively reduced the diarrhea occurrence in a dose-dependent manner (Fig. 4b), concomitant with the dose-dependent reductions of free IgE and MCPT-1 levels (Fig. 4c, d). Likewise, IgE$_{TRAP}$ dose-dependently inhibited the body temperature drop in a peanut-induced systemic anaphylaxis model (Supplementary Fig. 3).

Furthermore, *B. longum* administered with powdered mouse chow enhanced the ability of IgE$_{TRAP}$ to inhibit diarrhea occurrence and serum levels of free IgE and MCPT-1 (Fig. 5b, c, d). IgE$_{TRAP}$ with or without *B. longum* induced an increase in total IgE levels (Fig. 5c), which has also been reported from the omalizumab treatment[27]. The rise in total IgE levels upon omalizumab treatment has been attributed to the longer half-life of IgE bound to omalizumab compared a free IgE[27]. In this OVA-induced food allergy model, IgE$_{TRAP}$ increased OVA-specific IgE levels as well as total IgE (Supplementary Fig. 4).

The enhancement of IgE$_{TRAP}$ efficacy by *B. longum* was observed even with short-term oral administration, starting from a day after the induction of food allergy by intragastric administration of OVA (Fig. 4). 10 µg of IgE$_{TRAP}$ in combination with *B. longum* achieved a similar therapeutic effect as 100 µg IgE$_{TRAP}$ given alone (Fig. 4b). Although *B. longum* has previously been reported to reduce mast cell numbers via apoptosis and attenuate food allergy symptoms[13], the therapeutic effect of *B. longum* alone was weaker than that of IgE$_{TRAP}$ alone (Fig. 5). We also confirmed the efficacy of IgE$_{TRAP}$ and *B. longum* in a peanut-induced anaphylaxis model, where the combination of IgE$_{TRAP}$ and *B. longum* effectively inhibited body temperature drop, reduced anaphylaxis symptom scores, and lowered the free IgE and MCPT-1 levels in serum (Fig. 6). In this model, IgE$_{TRAP}$ increased the levels of both total IgE and peanut-specific IgE (Fig. 6d). Enhanced anti-allergic effect of IgE$_{TRAP}$ by the combined treatment with *B. longum* was prevalent even when the dose of *B. longum* administration was reduced by 5-fold from $5 \times 10^9$ cfu head$^{-1}$ to $1 \times 10^9$ cfu head$^{-1}$, and the duration of administration was reduced by 2-fold from 4 weeks to 2 weeks (Supplementary Fig. 5).

To evaluate whether the orally administered *B. longum* effectively colonized the intestines or induced meaningful changes in the intestinal microbial community, BALB/c littermates with OVA-induced food allergy were treated, or not, with *B. longum*. Then the feces and cecal contents were collected (Supplementary Fig. 6a) and microbial analysis was performed using quantitative PCR (qPCR), bacterial 16 S rRNA-targeted sequencing, and bacterial culturing. The results from the qPCR analysis showed that the relative abundance of *B. longum* in feces was highest at 4 h after the oral administration and its relatively high abundance was maintained for 12 h, but *B. longum* was hardly detected after 24 h (Supplementary Fig. 6b). Although the allergic reaction became more and more severe as the number of oral administration of OVA increased, the abundance of *B. longum* detected from fecal contents was similar between each cycle of OVA administration. (Supplementary Fig. 6b). In support of the qPCR results, *B. longum* was best cultured from the feces collected 4 h after the oral administration, indicating that the *B. longum* given to the mice were viable (Supplementary Fig. 6c and d). However, it could hardly be cultured from the feces collected one day later. Next, 16 S rRNA gene sequencing and the microbial community analysis was performed on the feces and cecal contents obtained 12 h after the last oral administration of *B. longum*. The oral administration of *B. longum* significantly increased the presence of *B. longum* in both the feces and cecal contents, but it did not significantly alter the microbial community (Supplementary Fig. 7 & Supplementary Table 2). Additionally, analysis of the microbial community in the feces collected prior to the oral administration of OVA and *B. longum* showed only minor differences in microbial communities between different littermates (Supplementary Fig. 8). In summary, these results indicate that *B. longum* does not easily colonize the host intestine and does not affect the changes in the intestinal microbial community, but it can enhance the therapeutic effect of IgE$_{TRAP}$.

### Administration of *B. longum* enhances IgE$_{TRAP}$-mediated suppression of mast cell numbers and goblet cell hyperplasia

IgE has been reported to increase mast cell numbers by enhancing cell survival[19,28,29], and we investigated whether the IgE$_{TRAP}$ in combination treatment with *B. longum* effectively suppresses this phenomenon in the small intestine of the OVA-induced food allergy model. IgE$_{TRAP}$ significantly reduced the number of mast cell numbers in the intestine, as shown by the reduced staining for chloroacetate esterase activity (Fig. 7a, b). The combination of IgE$_{TRAP}$ and *B. longum* induced an even further reduction in the number of mast cells. Moreover, IgE$_{TRAP}$ alone significantly reduced the number of basophils in the blood as well as the level of IgE bound to basophils, in addition to the reduction of the free IgE levels in serum (Supplementary Fig. 9). Basophils, which are crucial effector cells for allergic responses, highly express FcεRI like mast cells, but in contrast to mast cells which are more abundant in peripheral tissues, basophils are more abundant in blood. Flow cytometry analysis revealed that the combination therapy with IgE$_{TRAP}$ and *B. longum* effectively reduced the frequency of the mast cells in the small intestine 12 days after the final intragastric administration of OVA for the induction of food allergy (Fig. 7c, d). This effect was not seen with IgE$_{TRAP}$ alone.

In the model of OVA-induced food allergy, goblet cell hyperplasia was also observed upon the disease induction. The administration of IgE$_{TRAP}$ in combination with *B. longum* significantly reduced the goblet cell hyperplasia (Fig. 8a, b). IL-33 has been reported to exacerbate food allergy symptoms by increasing degranulation and survival of mast cells[30,31], and induce goblet cell differentiation and *MUC2* expression[32,33]. IgE$_{TRAP}$ significantly reduced *IL-33* expression, and this effect was greater when IgE$_{TRAP}$ and *B. longum* were given in combination (Fig. 8c). *MUC2* expression was also more effectively reduced by the combined administration with IgE$_{TRAP}$ and *B. longum* (Fig. 8d).

### Discussion

To generate a IgE$_{TRAP}$ as a therapeutic agent that blocks IgE from binding to IgE receptors, we linked the extracellular domain of FcεRI to an IgD/IgG4 hybrid Fc domain. With seven sites for N-glycosylation in

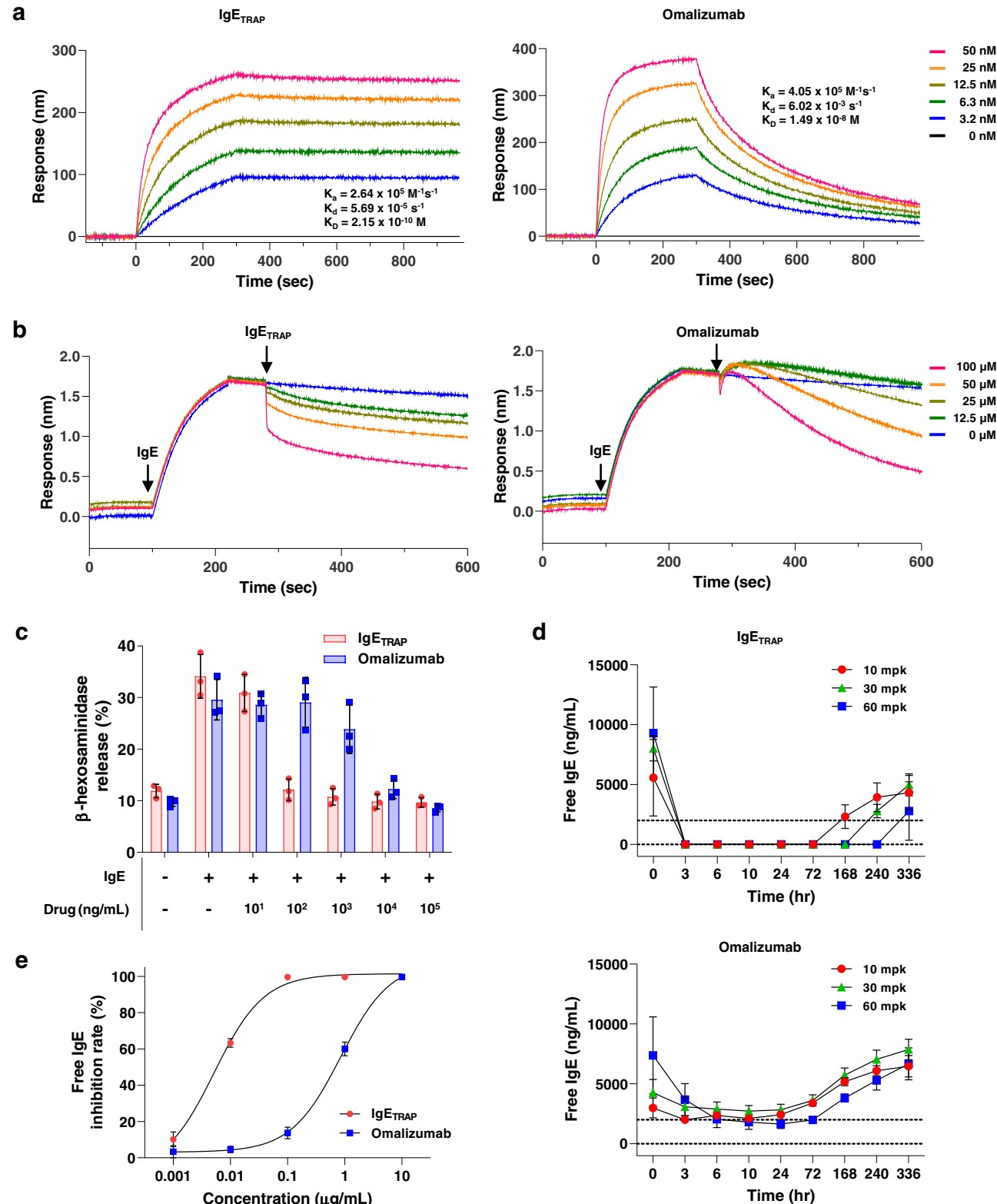

its FcεRIα extracellular domain[34] and one site in its IgD/IgG4 hybrid Fc domain, IgE$_{TRAP}$ is a heavily glycosylated protein. Although IgE$_{TRAP}$ retains the FcRn binding site to lengthen its half-life, its high glycan content may lead to rapid clearance in vivo through its interaction with glycan receptors. An attempt to generate FcεRI-Fc fusion approach has been explored in the past[35]. However, it failed to yield any drug products due to the issues of rapid clearance and potential immunogenicity. We sought to address the issue of rapid removal by capping

the glycan sites with sialic acid moieties[26]. Increasing the sialylation of the IgE$_{TRAP}$ suppressed its clearance while increasing its bioactivity in vivo (Table 2 and Fig. 3). In particular, IgE$_{TRAP}$ with sialic acid content higher than 14.9 mol mol$^{-1}$ was observed to be preferable as a therapeutic agent in the mouse model of IgE-mediated passive systemic anaphylaxis (Fig. 3d, e).

The total and antigen-specific IgE levels in serum were increased after the IgE$_{TRAP}$ treatment (Fig. 5c, Fig. 6d and Supplementary Fig. 4),

**Fig. 2 | Comparison of the inhibitory abilities of IgE$_{TRAP}$ and omalizumab on IgE binding to FcεRI and mast cell degranulation. a** The binding ability of IgE$_{TRAP}$ and omalizumab to human IgE. These experiments were analyzed by surface plasmon resonance (SPR). The figure is representative of two independent experiments with similar results. **b** Dissociation of IgE-FcεRI complex by IgE$_{TRAP}$ and omalizumab. After human IgE was bound to sensor-immobilized FcεRI, dissociations of IgE-FcεRI complexes by IgE$_{TRAP}$ and omalizumab were examined by bio-layer interferometry (BLI). **c** Suppression of mast cell degranulation by IgE$_{TRAP}$ (Red) and omalizumab (Blue). β-hexosaminidase release assays were performed in triplicate using the human mast cell line LAD-2 and the relative quantities of β-hexosaminidase activity were calculated. The figure is representative of five independent experiments. Data are presented as mean values ± SD. **d** Reduction of free IgE serum levels in cynomolgus monkeys after

the administration of IgE$_{TRAP}$ and omalizumab. 10 mg kg$^{-1}$ (mpk), 30 mpk, or 60 mpk IgE$_{TRAP}$ or omalizumab was administered subcutaneously with high basal serum IgE levels. Blood samples were collected at various time points after each drug administration and serum levels of free IgE, which has the ability to bind to IgE receptors on effector cells without binding to IgE inhibitors and can cause allergic reaction, were determined by ELISA ($n = 4$ animals in groups of 10 mpk and 30 mpk; 2 animals in 60 mpk group). Data are presented as mean values ± SEM. **e** Reduction of free IgE levels in the serum of CSU patients by IgE$_{TRAP}$ (Red) and omalizumab (Blue). Sera of CSU patients ($n = 6$) were incubated respectively with various concentrations (0, 0.001, 0.01, 0.1, 1 and 10 μg mL$^{-1}$) of IgE$_{TRAP}$ or omalizumab at 37 °C for 1 h and then free IgE level in each sample was determined by ELISA. Data are presented as mean values ± SD. Source data are provided as a Source Data file.

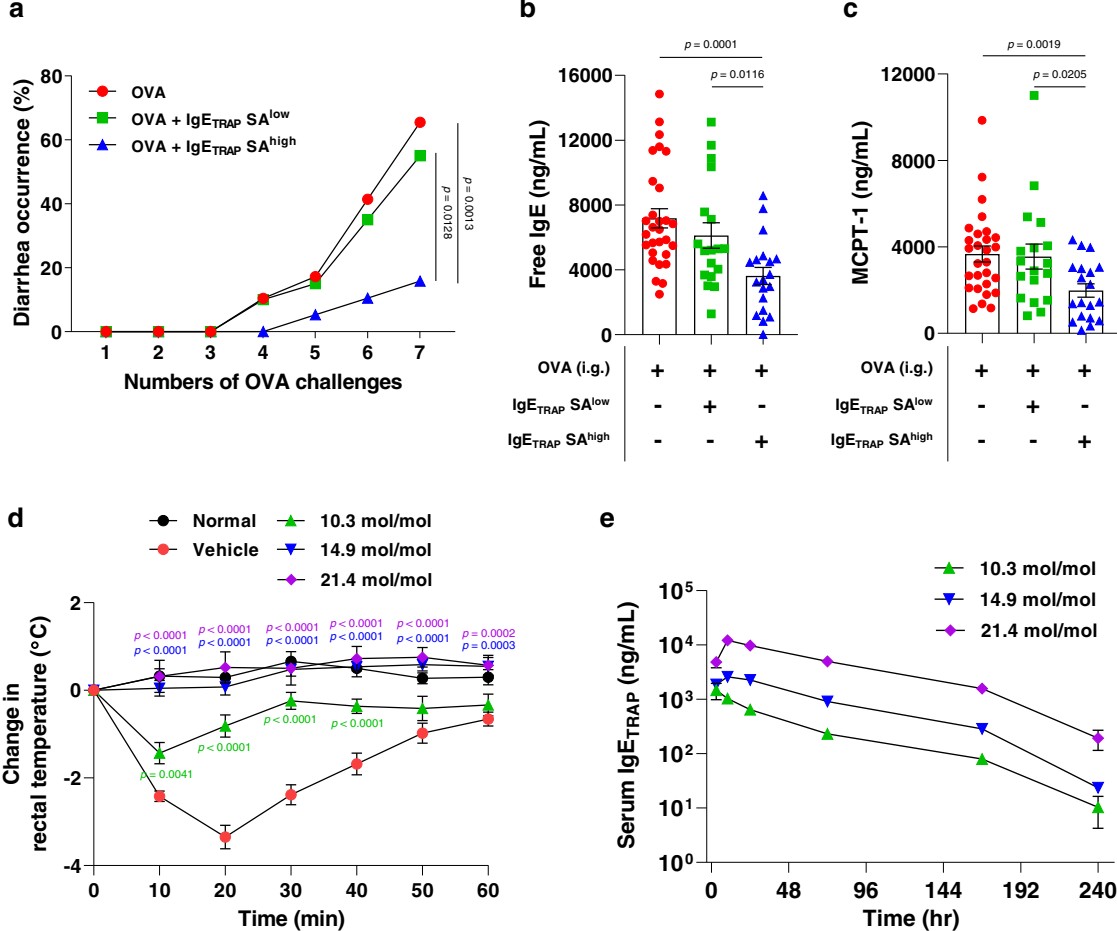

**Fig. 3 | Changes in therapeutic efficacy and in vivo serum concentrations of IgE$_{TRAP}$ according to sialic acid content. a–c** Therapeutic efficacy of IgE$_{TRAP}$ according to sialic acid content in the OVA-induced food allergy model that cause acute diarrhea. IgE$_{TRAP}$ SA$^{high}$ (19.7 mol mol$^{-1}$) or IgE$_{TRAP}$ SA$^{low}$ (7.7 mol mol$^{-1}$) were intraperitoneally injected into mice the day after the 2nd oral administration of OVA (50 mg). Therapeutic effect of IgE$_{TRAP}$ SA$^{high}$ and IgE$_{TRAP}$ SA$^{low}$ were assessed by suppression of diarrhea occurrence (**a**) and serum levels of free IgE (**b**) and MCPT-1 (**c**). The combined data of three independent experiments are shown ($n = 29$ mice in OVA group (Red circle); 18 mice in OVA + IgE$_{TRAP}$ SA$^{low}$ group (Green square); 19 mice in OVA + IgE$_{TRAP}$ SA$^{high}$ group (Blue triangle)). Statistical analyses were performed by Kaplan–Meier survival curve analysis with log-rank (Mantel-Cox) test to OVA + IgE$_{TRAP}$ SA$^{high}$ group (**a**) and unpaired two-tailed Student's $t$-test (**b**, **c**). **d** Therapeutic efficacy of IgE$_{TRAP}$ according to sialic acid content in the IgE passive systemic anaphylaxis model. IgE$_{TRAP}$ (10 mg kg$^{-1}$) with varying sialic acid content was

subcutaneously administered and rectal temperature was monitored to assess the therapeutic effect of IgE$_{TRAP}$ ($n = 5$ mice in Normal group (Black circle); 7 mice in Vehicle group (Red circle); 7 mice in groups of IgE$_{TRAP}$ with sialic acid contents, 10.3 mol mol$^{-1}$ (Green triangle), 14.9 mol mol$^{-1}$ (Blue triangle) and 21.4 mol mol$^{-1}$ (purple diamond). Statistical analysis was performed by Two-way repeated ANOVA with Dunnett's multiple comparison compared to vehicle group. **e** Changes in in vivo serum concentrations of IgE$_{TRAP}$ according to sialic acid content. IgE$_{TRAP}$ (10 mg kg$^{-1}$) with varying sialic acid content was subcutaneously injected into mice and IgE$_{TRAP}$ levels in serum were measured at various time points by ELISA ($n = 3$ mice in groups of IgE$_{TRAP}$ with sialic acid contents, 10.3 mol mol$^{-1}$ (Green triangle), 14.9 mol mol$^{-1}$ (Blue triangle) and 21.4 mol mol$^{-1}$ (purple diamond). All data are presented as mean values ± SEM. Specific $p$-values are indicated in the figure. Source data are provided as a Source Data file.

similar to what has been reported from the omalizumab treatment[27]. This phenomenon may be explained by IgE bound to $IgE_{TRAP}$ remaining in the body longer than free IgE, especially since both $IgE_{TRAP}$ and omalizumab have FcRn binding sites which are involved in half-life extension.

We evaluated the risk of immunogenicity of $IgE_{TRAP}$ using EpiScreen assays with $CD8^+$-depleted PBMCs from healthy human donors. $IgE_{TRAP}$ induced helper T cell responses in less than 10% of donors, comparable to the response rate observed from omalizumab and trastuzumab (Table 1, Supplementary Table 1, and Supplementary Fig. 1). Omalizumab and trastuzumab have been reported to have a very low clinical immunogenicity of 0.1%[36]. The comparable results between these antibodies and $IgE_{TRAP}$ from the Episcreen assay suggest that $IgE_{TRAP}$ may also have a minimal risk of immunogenicity. Moreover, high content of sialic acids on the $IgE_{TRAP}$ may further reduce the chance of anti-drug antibodies (ADA) production as the

sialic acids have been reported to induce B cell apoptosis by binding to Siglec-2[37].

High concentrations of antigen-specific IgG1 antibodies can induce passive systemic anaphylaxis in an antigen-rich environment via activation of low-affinity IgG receptors[38]. The IgG-mediated anaphylaxis is exacerbated in the absence of IgE signaling[3–7]. Since $IgE_{TRAP}$ completely lacks binding sites for FcγRs (Fig. 1), $IgE_{TRAP}$ is unlikely to induce the IgG-mediated anaphylaxis as seen with omalizumab. With a higher affinity for IgE than omalizumab, $IgE_{TRAP}$ better suppresses mast cell degranulation and free IgE serum levels (Fig. 2), demonstrating $IgE_{TRAP}$'s potential as a safer and more potent therapeutic option for treating IgE-mediated allergic diseases.

The high-affinity monoclonal anti-IgE antibody, ligelizumab (QGE031), has been in development as a next-generation IgE antibody and recently received FDA Breakthrough Therapy designation for CSU patients, but it has not yet to receive FDA approval. Ligelizumab inhibits IgE from binding to FcεRI and activating basophil more effectively than omalizumab[39]. Furthermore, ligelizumab enables IgE interaction with low affinity IgE receptor, CD23, whose interaction on B cells is known to suppress IgE synthesis[39]. Nevertheless, ligelizumab is an IgG1 antibody, like omalizumab, and therefore toxicities associated with IgG1, such as ADCC, CDC, and IgG-mediated anaphylaxis, may also apply to ligelizumab. Ligelizumab, unlike omalizumab, cannot dissociate complexes of IgE and FcεRI[39]. In contrast, we showed that $IgE_{TRAP}$ can dissociates complexes of IgE and FcεRI more rapidly and effectively than omalizumab (Fig. 2b). Given that sialic acid signaling has been reported to reduce IgE synthesis by inducing B cell apoptosis[37], high contents of sialic acid moieties on $IgE_{TRAP}$ may explain the higher potency of $IgE_{TRAP}$ compared to omalizumab.

**Table 2 | Pharmacokinetics profiles according to sialic acid content of $IgE_{TRAP}$**

| Sialic acid content (mol mol⁻¹) | $T_{1/2}$ (hr) | $T_{max}$ (hr) | $C_{max}$ (µg mL⁻¹) | $AUC_{last}$ (hr·µg mL⁻¹) | $AUC_{0-\infty}$ (hr·µg mL⁻¹) |
|---|---|---|---|---|---|
| 10.3 | 37.6 | 3 | 1.5 ± 0.5 | 62.0 ± 5.5 | 62.7 |
| 14.9 | 34.8 | 10 | 2.6 ± 0.2 | 197.3 ± 5.6 | 199.0 |
| 21.4 | 39.9 | 10 | 12.1 ± 0.7 | 953.9 ± 45.1 | 969.5 |

$T_{1/2}$, half-life; $T_{max}$, Time of maximum concentration observed; $C_{max}$, maximum concentration observed; $AUC_{last}$, area under the curve from time zero to time of last measurable concentration; $AUC_{0-\infty}$, area under the curve from time zero to infinity.

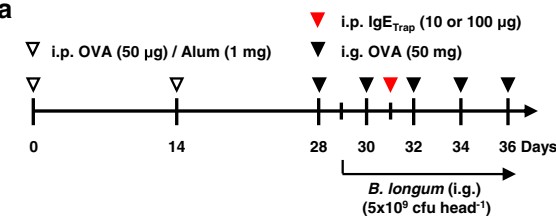

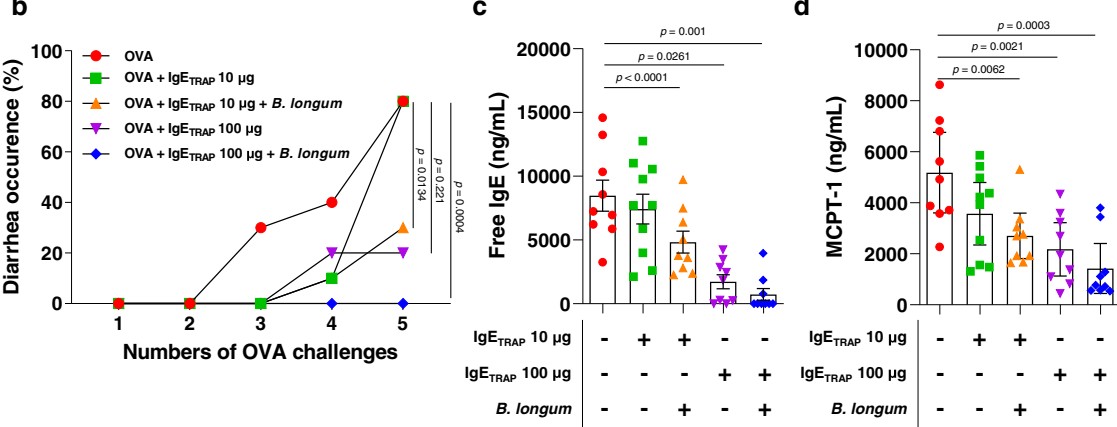

**Fig. 4 | Dose-dependent therapeutic effect of $IgE_{TRAP}$ in combination with *B. longum* in a food allergy model causing acute diarrhea. a** Experimental schedule for an OVA-induced food allergy model and administrations of $IgE_{TRAP}$ and *B. longum*. *B. longum* ($5 \times 10^9$ cfu head⁻¹) was administered every two or three days from the day after 1st oral administration of OVA. i.p., intraperitoneal; i.g., intragastric. **b** Suppression of allergy-induced diarrhea by $IgE_{TRAP}$ alone and $IgE_{TRAP}$ in combination with *B. longum*. The experiments were independently performed twice and a representative graph is shown. **c, d** free IgE (**c**) and

MCPT-1 (**d**) levels in serum ($n = 10$ mice in groups of OVA (Red circle), OVA + $IgE_{TRAP}$ 10 µg (Green square), OVA + $IgE_{TRAP}$ 10 µg + *B. longum* (Orange triangle), OVA + $IgE_{TRAP}$ 100 µg (Purple triangle), and OVA + $IgE_{TRAP}$ 100 µg + *B. longum* (Blue diamond)). Statistical analyses were performed by Kaplan–Meier survival curve analysis with log-rank (Mantel-Cox) test to OVA group (**b**) and unpaired two-tailed Student's *t*-test (**c, d**). All data are presented as means values ± SEM. Specific *p*-values are indicated in the figure. Source data are provided as a Source Data file.

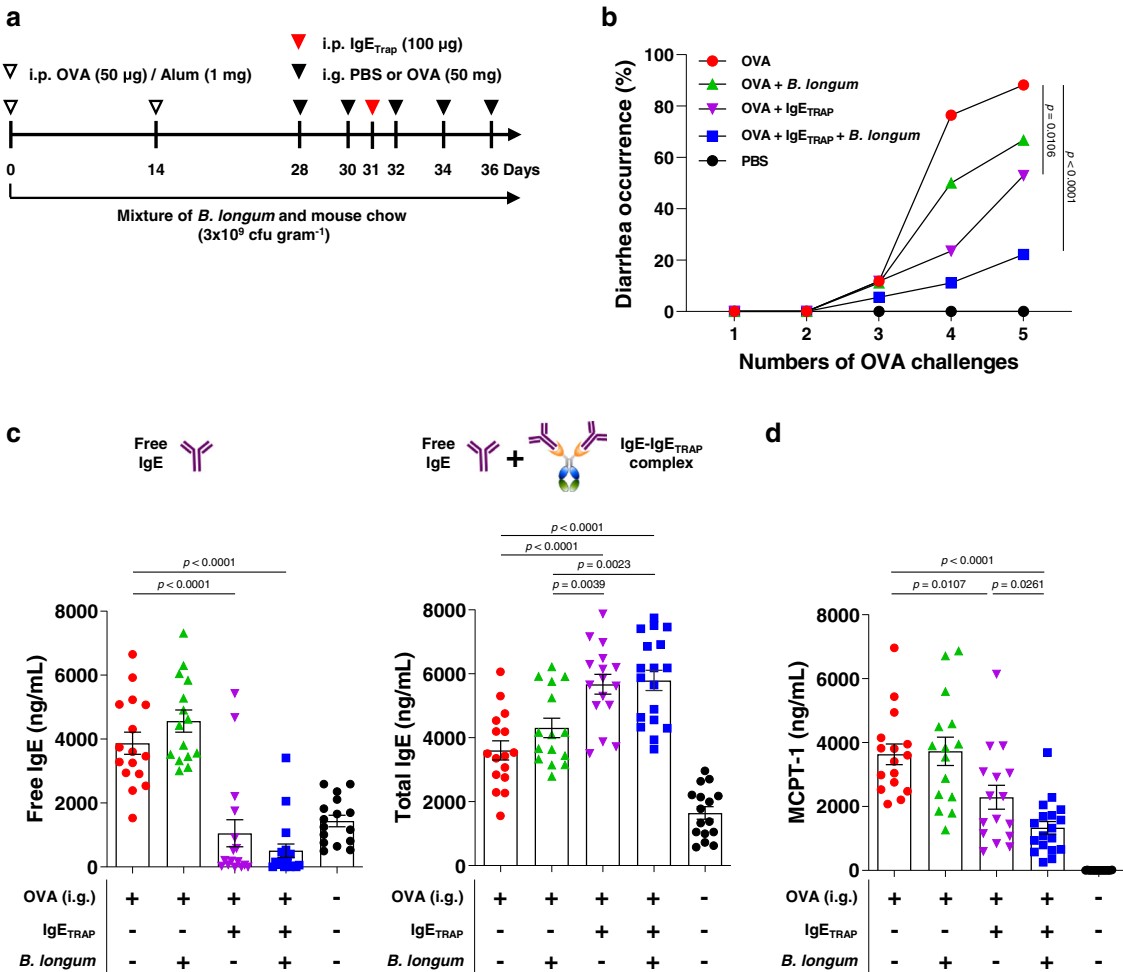

**Fig. 5 | Enhancement of therapeutic effects of IgE_TRAP by *B. longum* in a food allergy model causing acute diarrhea. a** Experimental schedule for an OVA-induced food allergy model and administrations of IgE_TRAP and *B. longum*. *B. longum* was mixed with powdered mouse food, which was offered on an ad-libitum basis to mice. i.p., intraperitoneal; i.g., intragastric. **b** Effect of IgE_TRAP and *B. longum* alone and in combination on allergy-induced diarrhea. **c, d** Serum levels of free and total IgE (**c**), and MCPT-1 (**d**). The combined data of three independent experiments is shown ($n = 16$ mice in groups of OVA (Red circle), OVA + IgE_TRAP + *B. longum* (Blue square) and PBS (Black circle); 15 mice in OVA + *B. longum* group (Green triangle); 18 mice in OVA + IgE_TRAP group (Purple triangle)). Statistical analyses were performed by Kaplan–Meier survival curve analysis with log-rank (Mantel-Cox) test to OVA group (**b**) and unpaired two-tailed Student's *t*-test (**c, d**). Data are presented as means values ± SEM. Specific *p*-values are indicated in the figure. Source data are provided as a Source Data file.

Reduction of mast cell number by IgE_TRAP was enhanced when *B. longum* treatment was given in parallel as a combination therapy (Fig. 7). Previously, we reported that *B. longum* lowers the mast cell number by inducing apoptosis[13]. The decrease in mast cell number by IgE_TRAP could be explained by the reduced IgE-mediated mast cell activation and survival[28,29]. Since mast cells produce IL-33 via IgE-mediated activation[40], the reduction in mast cell number could also result in the reduced IL-33 expression. In support of this idea, IgE_TRAP significantly reduced the mast cell number and the *IL-33* expression, and this effect was enhanced by *B. longum* (Fig. 7 and 8d). Through an autocrine positive feedback regulation, IL-33 may exacerbate food allergy by promoting mast cell survival and degranulation[30,31], and by inducing goblet cell hyperplasia both directly and indirectly[32,33]. IgE_TRAP in combination with *B. longum* was observed to effectively inhibit both the mast cell degranulation and the goblet cell hyperplasia (Figs. 7, 8a, b, and c).

*B. longum* has previously been reported to induce mast cell apoptosis and improve food allergy symptoms[13]. Although daily administration of *B. longum* was less effective than a single injection of 100 µg IgE_TRAP, the *B. longum* dramatically enhanced the therapeutic effects of IgE_TRAP (Fig. 5). Similar results were obtained even when *B. longum* was administered a day after the induction of food allergy by

intragastric administration of OVA, (Figs. 4, 6, and Supplementary Fig. 5), suggesting that the probiotic given after the onset of the disease can effectively enhance the therapeutic effect of IgE_TRAP in food allergy. Some gut bacteria can improve allergic disease symptoms by increasing Treg cell populations and reducing IgE and Th2 cytokine levels[41,42]. Probiotics other than *B. longum* may also enhance the therapeutic effect of IgE_TRAP through such mechanisms.

*B. longum* was hardly detected in fecal samples 24 hrs after the oral administration and had no effect on the intestinal microbial community, indicating that *B. longum* does not easily colonize the host gut when given as probiotics. Nevertheless, *B. longum* enhanced the therapeutic effect of IgE_TRAP even when it was administered after the induction of the allergy (Fig. 4). These observations suggest that therapeutics effect of *B. longum* is through a direct action of *B. longum* itself rather than an indirect action through changes in the intestinal microbial community. One possible mechanism of direction action by *B. longum* is its release of extracellular vesicle-derived protein which has been reported to penetrate the intestinal epithelial cells and selectively induce mast cell apoptosis[13] (Supplementary Fig. 10).

Although IgE blockade is a viable therapeutic approach for IgE-mediated allergies, the most widely prescribed treatment is omalizumab, whose efficacy is limited for patients with high IgE levels and

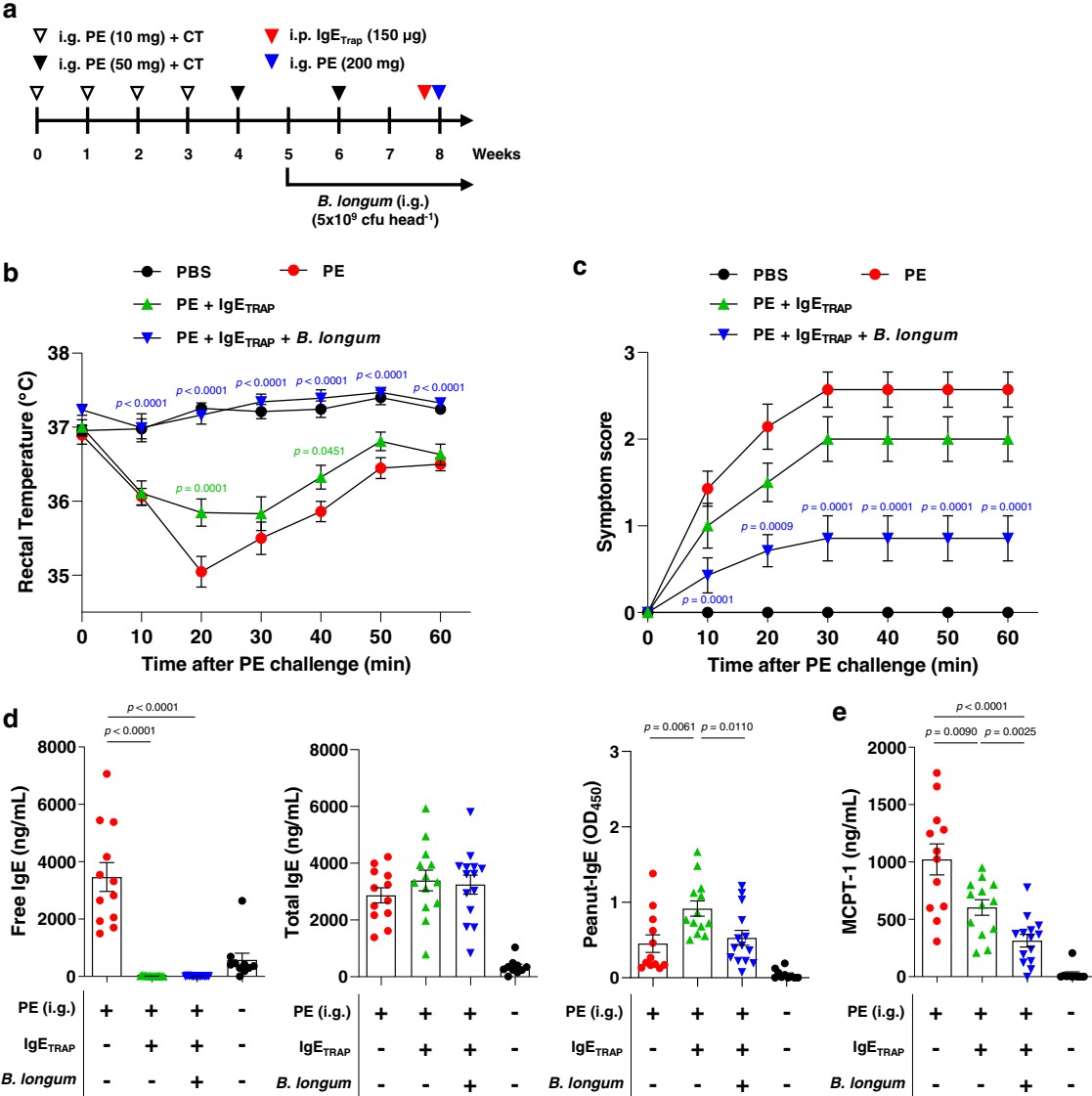

**Fig. 6 | Enhancement of therapeutic effects of IgE$_{TRAP}$ by *B. longum* in a peanut-induced systemic anaphylaxis model. a** Experimental schedule for a peanut-induced systemic anaphylaxis model and administrations of IgE$_{TRAP}$ and *B. longum*. *B. longum* ($5 \times 10^9$ cfu head$^{-1}$) was intragastrically administered every day from 5 weeks from the starting point of this anaphylaxis model. PE peanut extract, CT cholera toxin, i.p. intraperitoneal, i.g. intragastric. **b, c** Changes of rectal temperature (**b**) and anaphylactic symptom score (**c**) by IgE$_{TRAP}$ alone and IgE$_{TRAP}$ in combination with *B. longum*. Changes in rectal temperature was monitored and the data shown in (**b**–**e**) is combination of two independent experiments ($n = 13$ mice in PE group (Red circle); 14 mice in groups of PE + IgE$_{TRAP}$ (Green triangle) and PE + IgE$_{TRAP}$ + *B. longum* (Blue triangle); 10 mice in PBS group (Black circle). Changes in anaphylactic symptom score were also monitored and representative data is shown in (**c**). **d, e** Serum levels of free, total and peanut-specific IgE (**d**), and MCPT-1 (**e**). Statistical analyses were performed by Two-way repeated ANOVA with Dunnett's multiple comparison compared to PE group (**b, c**) and unpaired two-tailed Student's *t*-test (**d, e**). Data are presented as means values ± SEM. Specific *p*-values are indicated in the figure. Source data are provided as a Source Data file.

safety is suboptimal due to its risk of anaphylaxis. Ligelizumab, which is in development, is expected to share similar issues due to its IgG1 Fc region. Our findings suggest that IgE$_{TRAP}$ as a new candidate for the treatment of IgE-mediated allergies, and co-administration IgE$_{TRAP}$ with *B. longum* more effectively alleviates food allergy symptoms by reducing effector cell numbers, free IgE levels, and mast cell degranulation. These data show the combination therapy with IgE$_{TRAP}$ and *B. longum* can be a safer and more effective therapeutic option for allergic patients with high IgE levels.

## Methods
The research complied with all relevant ethical regulations. Animal experimental protocols were approved by the Institutional Animal Care and Use Committees (IACUC) of Genexine Inc., CN-Biologics Inc.,

GI-Biome Inc. and Seoul National University. Human samples were provided after obtaining informed consent and approval by the institutional review board (IRB) of Seoul National University Bundang Hospital.

### Mice
Six- to eight-week-old female BALB/c mice originated from Charles River Laboratories (USA) were purchased from ORIENT BIO Inc. in Korea, while 6-week-old female littermate BALB/c mice originated from Taconic Biosciences (USA) were purchased from Nara Biotech Inc. in Korea. 4-week-old female C3H/HeJ mice originated from University of Tokyo were purchased from Japan SLC Inc. Mice were supplied with normal diet chow (15 or 18% protein composition) and allowed to eat freely. Experiments were performed under specific pathogen-free (SPF)

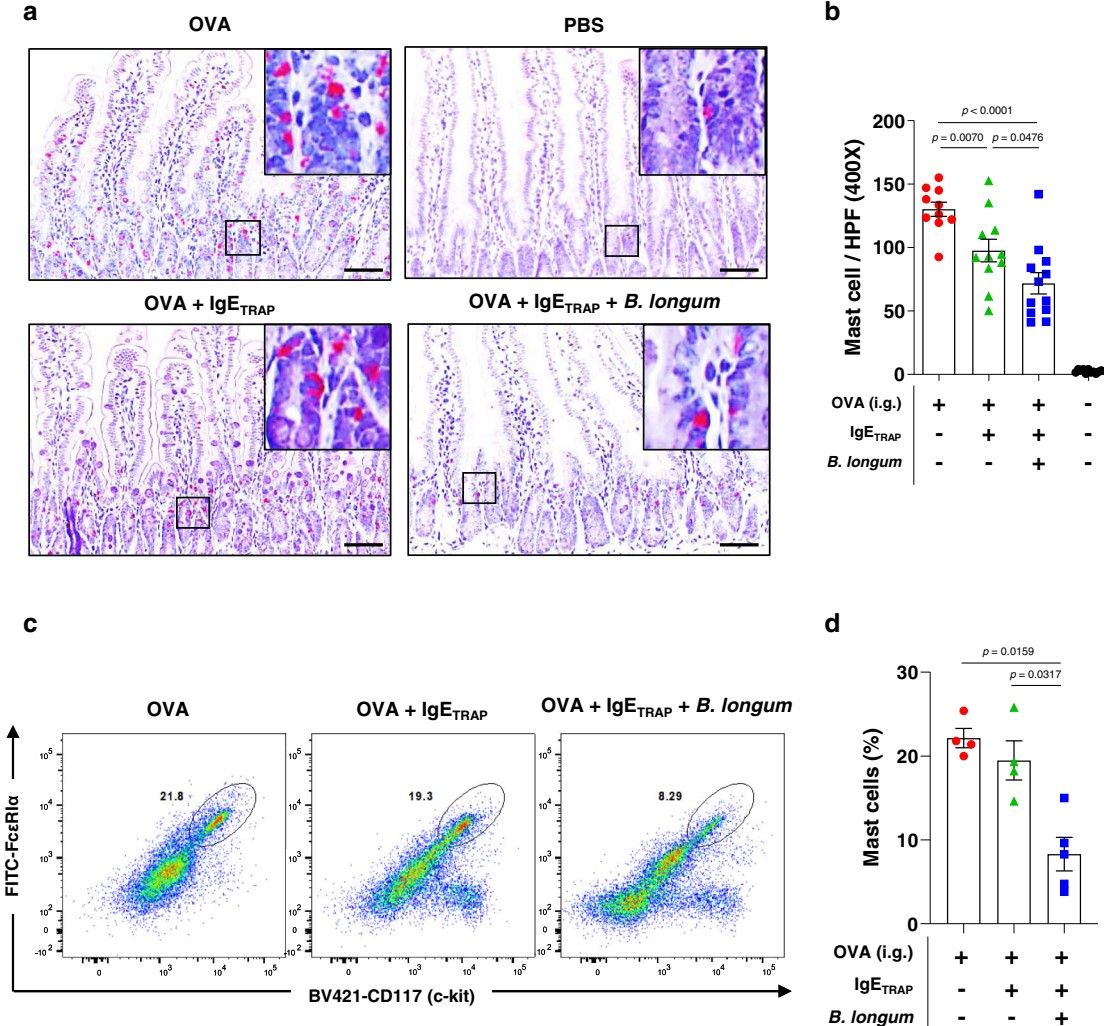

**Fig. 7 | Suppression of mast cell proliferation by IgE_TRAP in combination with *B. longum*. a**, **b** Histological analysis of the numerical changes of mast cells in the small intestine from the mice with OVA-induced food allergy after the intra-peritoneal injections of 100 μg of IgE_TRAP was intraperitoneally injected with or without *B. longum* given with food. Representative paraffin sections of the jejunum from each experimental group were stained for mast cells (red), which were shown in (**a**). Scale bar; 100 μm. Magnification of the intestine showing individual mast cells. Mast cells were counted in two different areas of the jejunum at 400x magnification (HPF; High power field) and average results are represented as a bar graph in (**b**). The combined data of two independent experiments is shown (*n* = 10 mice in OVA group (Red circle); 12 mice in OVA + IgE_TRAP group (Green triangle); 11 mice in groups of OVA + IgE_TRAP + *B. longum*

(Blue square) and PBS (Black circle)). **c**, **d** Flow cytometric analysis of the mast cells in the small intestine from the mice with OVA-induced food allergy after the intraperitoneal injections of 100 μg of IgE_TRAP with or without *B. longum* ($5 \times 10^9$ cfu head$^{-1}$) orally gavaged every two or three days from the day after 1st oral administration of OVA. Twelve days after the last OVA oral administration, the proportion of mast cells in the small intestine was analyzed by flow cytometry (*n* = 4 mice in groups of OVA (Red circle) and OVA + IgE_TRAP (Green triangle); 5 mice in OVA + IgE_TRAP + *B. longum* group (Blue square)). Representative flow cytometry data shown in (**c**) and all data are represented as a bar graph in (**d**). Statistical analysis was performed by unpaired two-tailed Student's *t*-test. Data are presented as means values ± SEM. Specific *p*-values are indicated in the figure. Source data are provided as a Source Data file.

or conventional conditions with housing environment of $21 \pm 3\,°C$ temperature, $50 \pm 10\%$ humidity and 12 h photoperiod. Each experimental protocols were approved by the IACUC of Genexine Inc., CN-Biologics Inc., GI-Biome Inc. and Seoul National University.

### Monkey study and human sample approval

The monkey study using 39- to 43-month-old female cynomolgus monkeys was approved by the IACUC of Korea Research Institute of Bioscience & Biotechnology. Serum samples from 6 CSU patients were collected with informed patient consent in accordance with procedures approved by IRB of Seoul National University Bundang Hospital.

### Cell line construction and purification of IgE_TRAP

IgE_TRAP was constructed by linking the C-terminus of the FcεRIα extracellular domain (25th−205th; 181 amino acids) to the N-terminus

of an IgD (133rd−170th; 38 amino acids)/IgG4 (121st−327th; 207 amino acids) hybrid Fc domain via a linker. The protein was expressed in dihydrofolate reductase-deficient CHO DG44 (Thermo Fischer, USA) cells and transferred with the α−2,6-sialyltransferase gene. IgE_TRAP was purified using a HiTrap rProtein A FF column (GE Healthcare, USA) and analyzed by SDS-PAGE under reducing and non-reducing conditions. Moreover, IgE_TRAP was further purified into two fractions according to the degree of its negative charge using a HiScreen Q Sepharose FF (GE Healthcare, USA) anion exchange chromatography column. The extent of sialylation of IgE_TRAP was confirmed in IEF gel or analyzed by HPLC (Waters, USA). Additionally, IgE_TRAP with superior ability to induce IgG1 Fc-mediated side effects was made by referring to the previous report[43,44]. This protein has a modified IgG1-Fc (T250Q/S298A/K334A/M428L) instead of the IgD/IgG4 hybrid Fc and is referred here as IgE_TRAP-IgG1M2.

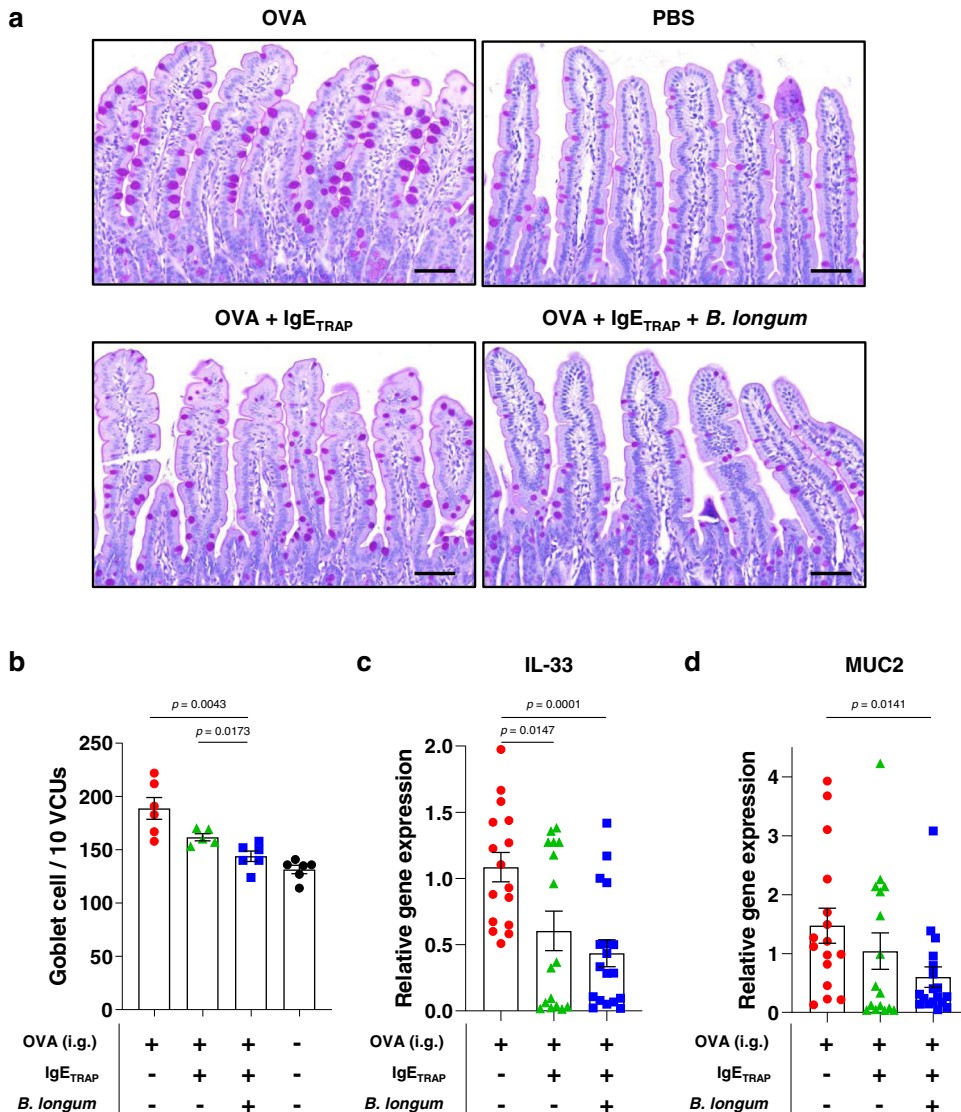

**Fig. 8 | Suppression of goblet cell hyperplasia and the expression of related genes by IgE_TRAP in combination with *B. longum*. a, b** Histological analysis of the numerical changes of goblet cells in the small intestine. Representative paraffin sections of the jejunum from each experimental group were stained for goblet cells (purple, 400x magnification). Goblet cells were counted in 10 randomly selected villus-crypt units (VCU) and the results represented as a bar graph. Scale bar; 100 μm. Representative result of three independent experiments (*n* = 6 mice in groups of OVA (Red circle), OVA + IgE_TRAP + *B. longum* (Blue square) and PBS (Black circle); 5 mice in OVA + IgE_TRAP group (Green triangle)). **c, d** Expression of IL-33 and

MUC2 genes in the small intestine. Relative expression of IL-33 (**c**) and MUC2 (**d**) was represented as a bar graph (*n* = 16 mice in groups of OVA (Red circle) and OVA + IgE_TRAP (Green triangle); 18 mice in OVA + IgE_TRAP + *B. longum* groups (Blue square)). All experiments were performed in an OVA-induced food allergy model. In this model, 100 μg of IgE_TRAP was intraperitoneally injected, and *B. longum* was mixed with powdered mouse food and was provided to mice. Statistical analysis was performed by unpaired two-tailed Student's *t*-test (**b**–**d**). Data are presented as means values ± SEM. Specific *p*-values are indicated in the figure. Source data are provided as a Source Data file.

## 3D structural modeling

The structural model of IgE_TRAP was manually designed using WinCoot and constructed with PyMOL software based on information available for FcεRIα (PDB accession 1F6A) and IgD/IgG4 Fc (PDB accession 1ADQ) in the Protein Databank.

## Surface plasmon resonance (SPR) assay

SPR assays were conducted using a ProteOn XPR36 (Bio-Rad, USA) device. For kinetic analysis of human IgE (Calbiochem, USA) binding omalizumab (Novartis, Swiss) and IgE_TRAP, 850 response units (RU) of omalizumab in acetate buffer (pH 5.5) and 500 RU of IgE_TRAP in acetate buffer (pH 4.0) were immobilized on a ProteOn™ GLC sensor chip (Bio-Rad, USA). Then, each dose of human IgE were interacted with immobilized IgE_TRAP or omalizumab. PBS containing 0.005% Tween-20

was used as the running buffer with a 30 μL min⁻¹ flow rate. Each data set was analyzed using the ProteOn manager software (Bio-Rad, USA).

## Bio-layer interferometry (BLI) assay

To assess the ability of IgE_TRAP and omalizumab to disrupt IgE-FcεRI complexes, His-tagged human FcεRI (SinoBiological, China; 5 μg mL⁻¹) was immobilized on Ni-NTA biosensors (Pall ForteBio, USA) for 600 s before human IgE (Abcam, UK; 100 nM) was allowed to bind sensor-immobilized FcεRI for 120 s. IgE_TRAP- and omalizumab-mediated dissociation of the IgE-FcεRI complexes was then measured for 600 s. To determine the binding potential of IgE_TRAP and omalizumab to IgG receptors, recombinant human FcγRI, FcγRIIA, FcγRIIB, FcγRIIIA, or FcγRIIIB (R&D Systems, USA; 5 μg mL⁻¹) in 10 mM acetate buffer (pH 5) was immobilized on activated AR2G biosensors for 300 s. The binding

kinetics of IgE$_{TRAP}$ and omalizumab to the sensor-immobilized IgG receptors were measured for 300 s. To assess Interaction of IgE$_{TRAP}$ and omaizumab to C1q protein, biotinylated IgE$_{TRAP}$ and omalizumab (10 µg mL$^{-1}$) were immobilized on streptavidin biosensors (Pall ForteBio, USA) and binding kinetics of C1q protein were analyzed[45]. All experiments were performed at 30 °C with the sample plate shaker speed set at 1000 rpm using an Octet RED384 or Octet K2 system (Pall ForteBio, USA).

### Antibody-dependent cell activation assay

To evaluate whether IgD/IgG4 hybrid Fc domain of IgE$_{TRAP}$ lacks IgG Fc-mediated function unlike omalizumab, ELISA plates were coated with Nitrophenyl-hapten (NP)-BSA (Bioresearch technologies, UK; 10 µg mL$^{-1}$) at 4 °C overnight and, after washing with PBS, incubated with anti-NP chimeric human IgE (JW8/1; GeneTax, USA; #GTX17414; 1:3 dilution) for 1 h at 37 °C. After washing with IMDM media containing 20% FBS, the plates were incubated with various concentrations of each drug (IgE$_{TRAP}$, IgE$_{TRAP}$-IgG1M2 or omalizumab) for 1 h at 37 °C. IgE$_{TRAP}$-IgG1M2, which is IgE$_{TRAP}$ with superior ability to induce IgG1 Fc-mediated side effects, was used as a positive control. The plates were cultured with NK103 cells (1 × 10$^5$ cells) expressing FcγRIIIA for 5 h at 37 °C in CO$_2$ incubator following a washing step with IMDM media containing 20% FBS. During the culture, NK cells were activated through FcγRIIIA to secrete granzyme B, which was used as an indicator of IgG-mediated function. Granzyme B in culture supernatant was measured with human granzyme B DuoSet ELISA kit (R&D systems, USA) according to the manufacturer's instruction. NK103 cells are FcγRIIIA-expressing NK cell line derived from human lymphoma, NK101[46]. NK103 was kindly provided by Dr. Sae Won Kim (SL-GIBEN Inc., Korea).

### EpiScreen assay

To examine the risk of immunogenicity, two independent EpiScreen assays were performed by Abzena Ltd. in UK. Abzena (Cambridge) Ltd is licensed (number 12627) by the Human Tissue Authority, the regulatory body for the Human Tissue Act (HTA) 2004. all activities that fall under this act were performed to HTA standards. PBMCs used in this experiment were obtained from at least 50 healthy human donors, who were selected to best represent the number and frequency of HLA-DR and HLA-DQ allotypes expressed in European/North American and the world population. PBMCs were depleted of CD8$^+$ T cells using CD8$^+$ RosetteSep (StemCell Technologies, Canada). T cells within PBMC populations were cultured for up to 8 days at 37 °C with 5% CO$_2$ in the presence of IgE$_{TRAP}$ (0.3 µM), omalizumab (0.3 µM; Novartis, Swiss), trastuzumab (0.3 µM; Genentech, USA), exenatide (5 µM; AstraZeneca, UK) or keyhole limpet hemocyanin (0.3 µM; KLH). Omalizumab and trastuzumab were used as a negative clinical control; exenatide as a clinical benchmark control; KLH as a neo-antigen control. On days 5, 6, 7 and 8, T cells were pulsed with 0.75 µCi [$^3$H]-Thymidine (Perkin Elmer, USA) and incubated for 18 h. T cells were harvested onto filter mats (Perkin Elmer, USA) and counts per minutes (cpm) were determined by MeltiLex scintillation, counting on a 1450 Microbeta Wallac Trilux Liquid Scintillation Counter (Perkin Elmer, USA). For data analysis, the stimulation index (SI) was calculated as follows: SI = mean of test wells (cpm)/baseline (cpm). A response was considered positive if the SI was equal to or greater than 1.9 (SI ≥ 1.9) and statically significant ($p < 0.05$) as compared to medium-only treated wells using the unpaired student's $t$-test. The percentage of donors that responded was calculated as follows: [the number of donors with a positive response over the entire time course (5–8 days)]/[the total number of the tested donors] x 100[36].

### β-hexosaminidase release assay

β-hexosaminidase release assays using LAD-2 cells were performed. LAD-2 cells were sensitized with biotinylated-IgE (100 ng mL$^{-1}$) before preincubation with IgE$_{TRAP}$ or omalizumab at different concentrations for 30 mins. The cells were then stimulated with streptavidin peroxidase (100 µg mL$^{-1}$). Total β-hexosaminidase was obtained by lysing of the LAD-2 cells in 0.1% Triton X-100. The supernatants were collected and incubated with an equal volume of 4 mM $p$-nitrophenyl $N$-acetyl-β-D-glucosaminide in citrate buffer for 1 h. The reactions were stopped by the addition of 0.4 M glycine buffer, and signals detected at a wavelength of 405 nm[47,48].

### ELISA

Free IgE levels in serum samples of monkeys and CSU patients were measured by ELSA[49]. ELISA plates were coated with IgE$_{TRAP}$ (1 µg mL$^{-1}$) and blocked with I-Block protein-based blocking reagent (Invitrogen, USA). Rabbit anti-human IgE antibody (RM122; Thermo Fischer, USA; #SA5-10201; 1:2000 dilution) and HRP-conjugated goat anti-rabbit IgG antibody (HP-6023; Novus biologicals, USA; #9190-05; 1:2000 dilution) were respectively used for IgE detection. 3,3′,5,5′-Tetramethylbenzidine (TMB) was used as the substrate for HRP enzyme. Absorbance was read at 450 nm and the concentrations were calculated using SoftMax Pro Software (Molecular Devices Corporation, USA). For the measurement of free IgE in mouse serum, the plates were coated with 1 µg mL$^{-1}$ IgE$_{TRAP}$ at 4 °C overnight and the rest of the assay was performed according to the manufacturer's protocols for the mouse total IgE ELISA kit (BioLegend, USA). For measurement of peanut-specific IgE levels in the mouse serum, ELISA plated were coated with peanut extract (25 µg mL$^{-1}$; GREER, USA) at 4 °C overnight and blocked with I-Block protein-based blocking reagent. Then, HRP-conjugated goat anti-mouse IgE antibody (Southern biotech, USA, #1110-05; 1:10000 dilution) was used for IgE detection. TMB was used as substrate of HRP enzyme. For measurement of OVA-specific IgE levels in the samples, Mouse Serum Anti-OVA IgE Antibody Assay Kit (Chondrex, USA) was used according to the manufacture's protocols. Concentrations of total IgE and MCPT-1 in the serum were measured with ELISA MAX™ Deluxe Set Mouse IgE (BioLegend, USA) and an MCPT-1 Mouse Uncoated ELISA Kit ELISA kit (Invitrogen, USA), according to the manufacturer's protocols.

### Pharmacokinetic analysis of IgE$_{TRAP}$

For pharmacokinetic analysis, IgE$_{TRAP}$ (10 mg kg$^{-1}$) with varying amount of sialic acid content was subcutaneously administered to BALB/c mice and blood samples were collected at various time points. For detection of IgE$_{TRAP}$ in mouse serum, plates were coated with anti-FcεR1α antibody (MAR-1; Invitrogen, USA; #14-5898-82; 0.5 µg mL$^{-1}$) and incubated with mouse serum samples after blocking with I-Block protein-based blocking solution. IgE$_{TRAP}$ was detected with an HRP-conjugated anti-human IgG4 pFc antibody (Southern Biotech, USA; #9190-05 1:10000 dilution) and TMB. Absorbance was read at 450 nm and the concentrations were calculated using SoftMax Pro Software (Molecular Devices Corporation, USA).

### Preparation of *B. longum*

*B. longum* used in this study was *B. longum* KACC 91563 isolated from feces of healthy Korean infants and was provided by the National Institute of Animal Science in Korea. In lab-scale, *B. longum* was anaerobically cultured in MRS broth (BD Biosciences, USA) containing 0.05% cysteine at 37 °C and were freeze-dried with protectant based on skim milk. Large-scale production was carried out by Mediogen Inc. in Korea. Cell counts of this freeze-dried *B. longum* was determined by dissolving in culture media with serial dilution and culturing on aga plates under an anaerobic condition.

### Food allergy model

For the OVA-induced food allergy model that features acute diarrhea, mice were intraperitoneally injected with 50 µg OVA (Grade V; Sigma–Aldrich, USA) and 1 mg of aluminum potassium sulfate

adjuvant (Sigma–Aldrich, USA) at day 0 and day 14. After day 28, 50 mg OVA (Grade III; Sigma–Aldrich, USA) was given as an oral challenge over a two-day interval. The animals were fasted for approximately 4 to 5 h prior to the OVA challenge and diarrhea occurrence was assessed by monitoring for up to 1 h after the challenge. IgE$_{TRAP}$ was intraperitoneally administered to the animals one day after the 2nd OVA oral challenge. Freeze-dried *B. longum* was mixed with powdered mouse chow at $3 \times 10^9$ cfu g$^{-1}$ for ad-libitum feeding and was freshly exchanged every 2 days. Considering that *B. longum* is very sensitive to contact with oxygen and one mouse eats an average of approximately 4 g of mouse chow per day, $3 \times 10^9$ cfu g$^{-1}$ was calculated to contain a sufficient amount of *B. longum* ($12 \times 10^9$ cfu in 4 g of powdered mouse chow) for a mouse to eat in one day. In some experiments, in order to orally administer the correct amount of *B. longum*, $5 \times 10^9$ cfu of *B. longum* dissolved in PBS was intragastrically administered to each mouse ($5 \times 10^9$ cfu head$^{-1}$) every 2–3 days.

For the peanut-induced systemic anaphylaxis model[50], C3H/HeJ mice were intragastrically sensitized with 10 mg peanut extract (GREER Laboratories, USA) and 20 µg cholera toxin (List Biological Laboratories, USA) 4 times weekly. The animals were then orally boosted with 50 mg peanut extract and 20 µg cholera toxin twice every 2 weeks. At week 8, 200 mg peanut extract was orally administered and rectal temperature was measured. In this model, IgE$_{TRAP}$ was intraperitoneally injected the day before challenge of 200 mg peanut, and $5 \times 10^9$ cfu of *B. longum* was intragastrically administered to each mouse ($5 \times 10^9$ cfu head$^{-1}$) every day from week 5 to 8. In some experiments, $5 \times 10^9$ cfu of *B. longum* dissolved in PBS was intragastrically administered to each mouse ($1 \times 10^9$ cfu head$^{-1}$) every day from week 6 to 8. Anaphylactic signs were scored as follows: 0, no signs; 1, rubbing and scratching around the snout and head; 2, puffiness around the eyes and snout, diarrhea, pilar erect, reduced activity, and/or decreased activity with increased respiratory rate; 3, wheezing, labored respiration, and cyanosis around the mouth and the tail; 4, no activity after prodding, or tremor and convulsions; 5, death.

For the IgE-mediated passive systemic anaphylaxis model[51], BALB/c mice were injected with 20 µg anti-dinitropheyl (DNP) IgE (SPE-7; Sigma–Aldrich, USA; #D8406; intraperitoneally) and 10 mg kg$^{-1}$ IgE$_{TRAP}$ (subcutaneously). After 24 h, 1 mg DNP-human serum albumin (HSA) (Sigma–Aldrich, USA) was intravenously injected and then rectal temperature was measured.

## Cell preparation and flow cytometry

Lamina propria (LP) cells were isolated from the small intestine as follows. After removal of fat tissues and Peyer's patches, the intestine was opened longitudinally, washed in PBS, and cut into 1–2 cm lengths. Epithelial cells were then removed by incubating the intestinal fragments in FACS buffer (PBS containing 3% FBS, 20 mM HEPES, 100 U mL$^{-1}$ penicillin, 100 µg mL$^{-1}$ streptomycin, 1 mM sodium pyruvate, and 10 mM EDTA) for 20 min at 37 °C with vigorous stirring. After washing with PBS, the intestinal fragments were minced and digested in RPMI-1640 media containing 3% FBS, 20 mM HEPES, 100 U mL$^{-1}$ penicillin, 100 µg mL$^{-1}$ streptomycin, 1 mM sodium pyruvate and 1 mM nonessential amino acids containing Collagenase D (Roche, Swiss; 400 U mL$^{-1}$) and DNase I (Roche, Swiss; 100 µg mL$^{-1}$) for 45 min 37 °C with continuous stirring. 40% Percoll (GE Healthcare, USA; 5 mL) solution was layered over cell suspension in 75% Percoll (5 mL) and centrifuged for 20 min at 2000 rpm at room temperature under brake-off conditions to collect the LP cells enriched at the interface. Cells were labelled with FITC-anti-FcεRIα (MAR-1; BioLegend, USA; #134306; 1:100 dilution) and Brilliant Violet 421-anti-c-kit (CD117) (ACK2; BioLegend, USA; #135124; 1:100 dilution) antibodies for flow cytometric analysis, where the mast cells were identified as double positive cells for FcεRIα and c-kit. For the analysis of basophils in blood, red blood cells in whole blood samples were lysed by RBC lysis buffer (Sigma, USA; #R7757). The remaining

cells were labelled with PE-anti-mouse IgE (RME-1; BioLegend, USA; #406908; 1:100 dilution) and FITC-anti-mouse CD49b (DX5; BioLegend, USA; #108906; 1:100 dilution) antibodies and basophils were identified as double positive cells for IgE and CD49b by flow cytometric analysis. All flow cytometry data were analyzed by a LSRFortessa flow cytometer (BD, USA) and were analyzed by FlowJo 10 software (Tree Star, USA).

## Histology

Jejunum tissue from the small intestine was fixed in 4% paraformaldehyde and embedded in paraffin to produce section slides. The slides were deparaffinized and stained for mast cells using a naphthol AS-D chloroacetate esterase kit (Sigma–Aldrich, USA) following the manufacturer's instructions. For goblet cells, slides were stained using a periodic acid-Schiff stain kit (ScyTek Lab, USA) according to the manufacturer's protocols. Images of stained slides were acquired on a Pannoramic MIDI (3D HISTECH Ltd, Hungary).

## Quantitative RT-PCR

Total RNA was extracted from the jejunum of the small intestine using TissueLyser II (Qiagen, USA) and RNeasy mini kit (Qiagen, USA), and cDNA was synthesized from total RNA of each sample using amfiRivert cDNA Synthesis Platinum Master Mix (GenDEPOT, USA). In PCR reactions, TB Green Premix Ex Taq (TAKARA, Japan) and the following primers were used: GAPDH (5′-GACAACTTTGGCATTGTGG-3′ and 5′-ATGCAGGGATGATGTTCTG-3′)[52], IL-33 (5′-ATGGGAAGAAGCTGATGGTG-3′ and 5′-CCGAGGACTTTTTGTGAAGG-3′)[53], MUC2 (5′-GCTGACGAGTGGTTGGTGAATG-3′ and 5′-GATGAGGTGGCAGACAGGAGAC3′)[54]. The relative expression of *IL-33* and *MUC2* were assessed using *GAPDH* as a reference.

## Fecal and cecal sample collection

Three littermates of 6 weeks old BALB/c mice were obtained from each of the 5 breeders and were evenly distributed to three groups and housed individually. To induce OVA food allergy, mice were intraperitoneally injected at day 0 and 14 with mixture of ovalbumin (OVA; 50 µg, Invivogen, USA) and aluminum hydroxide (Alum; 1 mg; Invivogen, USA), which had been confirmed to be free of lipopolysaccharide (LPS) contamination. Prior to oral administrations of OVA and *B. longum*, feces were collected from each mouse. From day 28, PBS or 50 mg OVA was intragastrically challenged at 1 PM every other day and was administered a total of 5 times. PBS or *B. longum* dissolved in PBS ($5 \times 10^9$ cfu head$^{-1}$) was intragastrically administered at every 8 AM from day 29. Fecal samples were obtained 4, 8, 12 and 24 h after the oral administrations of *B. longum* on day 29, 31, 33, and 35. Feces and cecal contents were collected 12 h after the last oral administrations of *B. longum*.

## *B. longum* culture from fecal samples

After the third oral administration of OVA in OVA-induced food allergy model, fecal samples were collected and weighed 4 h after *B. longum* administration. The anaerobic culture chamber (Coy Laboratory Products Inc., USA) used mixed gas (5% H$_2$, 5% CO$_2$, and 90% N$_2$) to remove O$_2$. For bacterial colony identification, colony-PCR and Sanger sequencing (BIONICS, Korea) were conducted using 16 S fecal samples were serially diluted and cultured on Bifidus Selective Medium (BSM) (Sigma-Aldrich, USA) agar plates under the anaerobic condition at 37 °C for 3 days. The rRNA gene-targeted universal primers (F: 5′-AGAGTTTGATCMTGGCTCAG-3′ and R: 5′-TACGGYTACCTTGTTACGACTT-3′). Identity verification was done using 16S-based ID App of EzBioCloud[55]. According to the product description, *Bifidobacterium* colonies, unlike other bacterial colonies, grow larger and are cream colored with dark brown spot in the center. Colonies from the randomly identified *B. longum*-treated samples were in similar shape with a dark brown spot and were confirmed to be *B. longum*. The quantity of *B. longum* per 1 g of

feces was calculated by counting the number of *Bifidobacterium* colony and multiplying the dilution factor.

### Relative quantification of *B. longum* in fecal samples

Fecal samples were suspended in 500 μL PBS and stored at −80 °C deep freezer until use. DNA was extracted from fecal samples by using FastDNA SPIN Kit for soil (MP Biomedicals, USA) according to the manufacturer's instructions and the amount of DNA was measured using NanoDrop 8000 spectrophotometer (Thermo Fisher, USA). The PCR reaction was run for a total of 40 cycles in QuantStudioTM 3 Real-Time PCR Instrument (Applied biosystems, USA) using 2X greenstar qPCR Master Mix with ROX Dye (Bioneer, Korea) and the following primers: *B. longum* specific primers (F: 5′-CGGCGTYGTGACCGTT GAAGAC-3′ and R: 5′-TGYTTCGCCRTCGACGTCCTCA-3′)[56] and bacterial common 16 S rRNA gene primers (F: 5′-AGAGTTTGATCM TGGCTCAG-3′ and R: 5′-TACGGYTACCTTGTTACGACTT-3′)[57]. The relative amount of *B. longum* among total bacteria was calculated by using a formula of $2^{[Ct (total bacteria) - Ct (B. longum) + 2]}$, where Ct is the value of cycle threshold.

### Intestinal microbial community analysis

Genomic DNA was extracted from cecal and fecal samples with FastDNA SPIN Kit for Soil (MP Biomedical, USA). PCR amplification was performed using 341 F/805 R primer set from V3 to V4 regions of the 16 S rRNA gene. Amplified products were purified using AMPure XP beads (Beckman Coulter, USA). Purified amplicons were pooled and sequenced on Illumina MiSeq (Illumina, USA) using a sequencing platform at Macrogen Inc. (Korea) according to the manufacturer's instructions. Raw paired reads were denoised and assembled using the DADA2 package[58]. Artificial sequences in each read were removed using the Skewer program[59]. Chimeric and contaminated reads containing small subunit rRNA gene sequences that originated from chloroplast, mitochondria, eukaryote, or archaea were removed using the Mothur pipeline v.1.43.0[60]. Cleaned high-quality reads were taxonomically assigned using the classify.seqs function in Mothur Pipeline with SILVA database[61]. The abundance of observed operational taxonomic units (OTUs) were evaluated after clustering the cleaned high-quality reads with 3% dissimilarity threshold using the OptiClust algorithm[62]. Weighted and unweighted-UniFrac distances among the samples were calculated with the sequences and abundance values of representative reads to cluster and compare the samples[63]. To normalize the sequence counts among the samples in distance calculation, randomly subsampled reads (150,000 reads), which were smaller than the minimum sample size, were used for microbial community analysis. One-way ANOSIM (analysis of similarities) test based on the UniFrac distance values was performed by using ANOSIM function in Mothur package with 10,000 permutations. The bacterial diversity values were calculated using estimators in Mothur and statistically tested by the Kruskal–Wallis test in R language (V.4.0.4) considering the dependency of the microbiome profiles on littermate groups. Multivariable association test was done through the generalized linear regression between taxonomic relative abundance and *B. longum* treatment by controlling littermates as a confounding variable. MaAsLin2 (Microbiome Multivariable Association with Linear Models) program[64] was used with the AST (Arc-sine transformation) options on microbial abundance. The diversity was visualized using ggplot2 package (V.3.3.5) and ggpubr package (V.0.4.0) in R language.

### Statistics and reproducibility

Statistical analysis of all data was conducted using GraphPad Prism software (v9.3.1) (GraphPad Software Inc.) and described in the figure legends. No statistical method was used to predetermine sample size. All the experiments were performed in a randomized manner. In some animal experiments, one or two samples with extreme deviations from the same group were excluded from data analysis. Diarrhea occurrence after allergen challenge was assessed by blinded observers. In other experiments, the investigators were not blinded since the randomization minimized their influence on experiments. All the experiments were repeated twice or more, and reproducibility was confirmed. Pharmacokinetic parameters were determined by sparse non-compartmental approach using the Phoenix WinNonlin program (version 8.1).

### Reporting summary

Further information on research design is available in the Nature Research Reporting Summary linked to this article.

## Data availability

The structure of IgE$_{TRAP}$ was based on the Protein Data Bank (PDB accession 1F6A and 1ADQ). The 16 s rRNA analysis of fecal and cecal samples data generated in this study have been deposited in the National Center for Biotechnology Information (NCBI) database under accession code PRJNA853889. All data generated in this study are provided in the published article and its Supplementary Information/Source Data file. Source data are provided with this paper.

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

## Acknowledgements

This study was supported by a grant of the Korea Health Technology R&D Project (Project No. HR16C0001) (H.-S. Park) through the Korea Health Industry Development Institute, funded by the Ministry of Health & Welfare, Republic of Korea, and a Cooperative Research Program for Agriculture Science and Technology Development grant from Rural Development Administration, Republic of Korea (Project No. PJ01331901, PJ01131601) (H.-T. Jin). We thank Lee Farrand and Dr. Yaein Amy Shim for proofreading of the manuscript.

## Author contributions

B.-G.Y., Y.C.S. and M.H.J. designed the experiments. S.B.A., B.-G.Y., G.J., D.-Y.K., S.-M.O., Y.-J.S., H.-W.H., D.L., C.-P.H., J.H.K. (Jung Hwan Kim), H.-T.J. and S.-W.L. performed animal experiments and related data analysis. S.B.A., J.K. (Jiyoung Kim), N.O., S.L., J.-Y.M., J.-A.K., J.-H.K. (Ji-Hyun Kim), J.K. (Jisoo Kim), K.L., M.S.R., S.-W.L., Y.-S.C. and H.-S.P. performed in vitro experiments and related data analysis. H.S.L. performed 3D structure modeling. M.-J.K., B.K.K. and Y.-K.P. cultured and analyzed gut microbiota. S.B.A., B.-G.Y. and M.H.J. wrote the manuscript.

## Competing interests

Y.C.S. and M.H.J. are inventors on patents related to this work (PCT/KR2019/000270 & PCT/KR2019/000524). Authors affiliated with GI Innovation Inc., GI Biome Inc., GI Cell Inc. and ProGen Inc. are employees of the respective companies, which are involved in the co-development of IgE$_{TRAP}$ and *Bifidobacterium longum* combination therapy for IgE-mediated food allergy. Y.-S.C. and H.-S.P. are consultants for GI Innovation. The remaining authors declare no competing financial and non-financial interests.

## Additional information

[1]Division of Integrative Biosciences and Biotechnology, Pohang University of Science and Technology (POSTECH), Pohang, Gyeongbuk, Republic of Korea. [2]Department of Life Sciences, Pohang University of Science and Technology (POSTECH), Pohang, Gyeongbuk, Republic of Korea. [3]Research Institute, GI Innovation Inc., Songpa, Seoul, Republic of Korea. [4]Research Institute, GI Biome Inc., Seongnam, Gyeonggi-do, Republic of Korea. [5]Department of Allergy and Clinical Immunology, Ajou University School of Medicine, Suwon, Republic of Korea. [6]Research Institute, GI Cell Inc., Seongnam, Gyeonggi-do, Republic of Korea. [7]Research Institute, ProGen Inc., Korea Bio Park, Seongnam, Gyeonggi-do, Republic of Korea. [8]Department of Internal Medicine, Seoul National University Bundang Hospital, Seoul National University College of Medicine, Seongnam, Republic of Korea. [9]World Premier International Immunology Frontier Research Center, Osaka University, Suita, Japan. [10]These authors contributed equally: Seong Beom An, Bo-Gie Yang. ✉e-mail: yangbg@gi-biome.com; ycsung@postech.ac.kr; jangmh@gi-innovation.com

