## [Peer Review File · Nature Communications]

Combined IgE neutralization and Bifidobacterium longum supplementation reduces the allergic responses in food allergy modelsREVIEWER COMMENTS

Reviewer #1 (Remarks to the Author):

The IgE inhibitor omalizumab is emerging as a treatment for allergic disease. It has FDA approval for severe asthma and chronic urticaria and is in clinical trials for other diseases, including food allergy. In this interesting paper the authors have created a new drug that aims to overcome some of the limitations of omalizumab. Their IgETRAP combines the high affinity human Fcepsilon R1 domain with an IgD/IgG4 hybrid Fc domain. IgETRAP cannot therefore bind to low affinity IgG receptors and would these not be expected to participate in ADCC or IgG-mediated anaphylaxis by binding to complement or receptors for IgG. The authors propose that IgETRAP will be both safer and more effective than omalizumab. Much of the data presented is compelling. Some concerns are outlined below.

The authors claim that “IgETRAP did not elicit a positive helper T cell response in EpiScreen assays using CD8+ depleted PBMCs from 50 healthy donors” but data is not shown. Please show this data. They say that in the same assay omalizumab “reportedly” elicits a positive response – why not test IgETRAP and omalizumab side-by-side?

It is not clear how to interpret the data in the monkeys since they are apparently not allergic – what is the clinical relevance of blocking serum IgE in a non-allergic individual?

The OVA/alum induced allergic diarrhea model is not a good model for systemic anaphylactic responses. Omalizumab is currently in clinical trials for peanut allergy in combination with oral immunotherapy. Repeating the experiments shown in a model of systemic anaphylaxis to food allergens (e.g. the Sampson PN/cholera toxin model) would have more clinical relevance.

Critically missing from the model presented in Fig. 3 is a measurement in the reduction of antigen (OVA) specific IgE. “Free” IgE as labeled in Fig. 3 apparently refers to total serum IgE. Are the OVA preparations used in the food allergy models LPS free? Typically Grade V OVA purchased from Sigma is significantly contaminated with LPS.

The authors treated some mice with IgETRAP plus freeze dried *B. longum*. Did *B. longum* colonize the mice? Were the authors able to recover live *B. longum* in feces from the treated mice? They have not done any experiment to validate the viability (or integrity) of their *B. longum* preparation. Importantly, the experiments are not littermate controlled. This makes it difficult to interpret the influence of administered bacteria since the authors have not controlled for differences in the composition of the microbiomes among the experimental groups (see Stappenbeck & Virgin “Accounting for reciprocal host-microbiome interactions in experimental science” *Nature* 2016, 534: 191).

Reviewer #2 (Remarks to the Author):

An and colleagues present an antibody fusion molecule which combines the alpha chain of the high affinity IgE Fc receptor with an Fc domain comprising an IgD and IgG4 fusion. This molecule, alone or in combination with the probiotic bacterium *Bifidobacterium longum* was tested as a treatment for allergy in vitro and in vivo. This work is interesting, there are novel

aspects to the study and the data show therapeutic promise in the systems tested. However, there are elements to the presentation of the study which do not allow for sufficient understanding or appraisal of the data and these need to be addressed.

- The choice of IgD domain in the Fc part of the construct and the combination with IgG4 domains is unclear and not justified anywhere in the manuscript. Why not use a full IgG4 Fc instead?

- The schematic in Figure 1 is not sufficient to convince the reader that the Fc construct is designed to minimise the binding to Fc gamma Receptors. IgG4 Fc domains bind very well to Fc gamma Receptors and unless specific mutations are introduced the domain will still bind to Fc gamma Receptors including the high affinity Fc gamma RI. Are there mutations introduced in the construct, where and why? These should be explained in the manuscript and represented in the schematic in Figure 1A and 1B. A clear explanation should be included as to how the construct is designed to avoid binding to Fc gamma Receptors.

- In the mast cell degranulation model:

- o LAD-2 cells are poor degranulators, to convince the reader that these cells are triggered by IgE immune complexes in the first place, the % degranulation of these cells should be shown as controls in the in vitro model and the % degranulation should be compared with the blocking by omalizumab and by IgE(TRAP) at different concentrations of each drug.

- o The data should be reproduced in rat basophilic leukemia cells.

- In the introduction is it incorrectly stated that Omalizumab is the only anti-IgE agent approved for clinical use. This is incorrect. The authors should acknowledge that Ligelizumab has also been approved and this antibody has even higher affinity for the Cepsilon3 domain of IgE. This is also a humanized monoclonal IgG1k and would be expected to suffer from the same issues as Omalizumab. This should be acknowledged, discussed, and explained in the manuscript. No comparisons are made here between IgE(TRAP) and this very high affinity antibody, an agent like IgE(TRAP) is likely to be compared with Ligelizumab in future because of the high affinity to IgE

- The authors claim that IgE(TRAP) did not elicit a positive helper T cell response in CD8+ EpiScreen assays using depleted peripheral blood mononuclear cells (PBMCs) from 50 healthy human donors, indicating that IgE(TRAP) exhibits minimal immunogenicity. Given that lack of immune activation is a big claim in this study, the experimental design should be included in the methods section and the data should be shown.

- The data presented in cynomolgus monkeys are not justified sufficiently. Monkeys have normal levels of IgE in the blood and are not an allergy model. The authors claim that this drug could work in instances of high serum IgE levels in allergic patients being the competitive advantage of this new construct. Therefore, this monkey model will not be representative of a patient situation.

An assay should be introduced to show lack of Fc-mediated functions of the construct, e.g. CDC or ADCC.

- The data showing differences in half-life of the constructs with high and low sialylation should be shown and explained in the discussion. Do the authors expect one of these variants to be a better therapeutic agent and why?

- What is the effect of the drug on human blood basophils?

- The Abstract should include a justification for combination treatment with the probiotic bacterium. As written this combination is not clear to the general readership.

Reponses to the reviewer's comments for the manuscript entitled: "Combination therapy with *IgE_{TRAP}* and probiotics effectively reduces IgE-mediated responses in an allergy model"

REVIEWER COMMENTS

Reviewer #1 (Remarks to the Author):

The IgE inhibitor omalizumab is emerging as a treatment for allergic disease. It has FDA approval for severe asthma and chronic urticaria and is in clinical trials for other diseases, including food allergy. In this interesting paper the authors have created a new drug that aims to overcome some of the limitations of omalizumab. Their IgETRAP combines the high affinity human Fcepsilon R1 domain with an IgD/IgG4 hybrid Fc domain. IgETRAP cannot therefore bind to low affinity IgG receptors and would these not be expected to participate in ADCC or IgG-mediated anaphylaxis by binding to complement or receptors for IgG. The authors propose that IgETRAP will be both safer and more effective than omalizumab. Much of the data presented is compelling. Some concerns are outlined below.

The authors claim that "IgETRAP did not elicit a positive helper T cell response in EpiScreen assays using CD8+ depleted PBMCs from 50 healthy donors" but data is not shown. Please show this data. They say that in the same assay omalizumab "reportedly" elicits a positive response – why not test IgETRAP and omalizumab side-by-side?

→ In accordance with the reviewer's comment, we have added the data from the EpiScreen assay in **revised Table 1**. According to a previous report (*PLoS One* 2016;11:e0159328), both trastuzumab (Herceptin) and omalizumab (Xolair) can be used as negative clinical controls in the EpiScreen assay, but we chose trastuzumab (Herceptin) because it appears to have a slightly weaker response than omalizumab (Xolair). The underlying data (*PLoS One* 2016;11: e0159328) is shown below. Furthermore, for clarity, we have added more descriptions of omalizumab in the **revised lines 240-243**.

Table 1 | EpiScreen assay

Sample	Mean SI*	SD	% Response
IgE _{TRAP}	N/A	N/A	0
PBS	N/A	N/A	0
Trastuzumab (Herceptin)	2.35	± 0.43	4
Exenatide (Bydureon)	2.47	± 0.64	14
Keyhole limpet hemocyanin (KLH)	9.15	± 8.75	90

*Stimulation Index (SI) = mean value of test wells (cpm) / baseline (cpm). The mean SI was calculated from the average of all positive donor responses observed during the entire time course (days 5-8). N/A indicates no data available.

It is not clear how to interpret the data in the monkeys since they are apparently not allergic – what is the clinical relevance of blocking serum IgE in a non-allergic individual?

→ We agree with the reviewer’s comment. In order to avoid confusion, we have clearly described the reason for using non-allergic monkeys in the **revised Lines 143-145**.

The OVA/alum induced allergic diarrhea model is not a good model for systemic anaphylactic responses. Omalizumab is currently in clinical trials for peanut allergy in combination with oral immunotherapy. Repeating the experiments shown in a model of systemic anaphylaxis to food allergens (e.g. the Sampson PN/cholera toxin model) would have more clinical relevance.

→ According to the reviewer’s comment, we have performed experiments using the peanut/cholera toxin model and obtained similar results to those of the OVA/Alum model. The data is shown in the **revised Figure 6** and **Supplemental Figure 1**, which been explained in **revised Lines 178-179, 197-199 and 382-391**.

Figure 6

Supplemental Figure 1

Critically missing from the model presented in Fig. 3 is a measurement in the reduction of antigen (OVA) specific IgE. "Free" IgE as labeled in Fig. 3 apparently refers to total serum IgE. Are the OVA preparations used in the food allergy models LPS free? Typically Grade V OVA purchased from Sigma is significantly contaminated with LPS.

→ Because IgE_{TRAP} inhibits degranulation of effector cells (e.g. mast cells and basophils) through the blocking of IgE binding to FcεRIα to alleviate allergy symptoms, a single injection of IgE_{TRAP} after induction of allergy responses cannot reduce the amount of total or antigen-specific IgE. Moreover, according to a previous report (*J Allergy Clin Immunol* 2016;137:507-16; Fig 2), *B. longum* alleviates food allergy symptoms by inducing mast cell apoptosis without reducing OVA-specific IgE. For these reasons, we expect that there will be no change in OVA-specific IgE levels in the experiments of Fig 3. In practice, combination therapy with IgE_{TRAP} and *B. longum* effectively reduce free IgE levels, but not total IgE levels as shown.

→ As per the reviewer's comment, the OVA we used was significantly contaminated with LPS. Therefore, in experiments using the peanut/cholera toxin model, we excluded the possibility of LPS-induced effects by using TLR4-defective C3H/HeJ mice.

The authors treated some mice with IgE_{TRAP} plus freeze dried *B. longum*. Did *B. longum* colonize the mice? Were the authors able to recover live *B. longum* in feces from the treated mice? They have not done any experiment to validate the viability (or integrity) of their *B. longum* preparation. Importantly, the experiments are not littermate controlled. This makes it difficult to interpret the influence of administered bacteria since the authors have not controlled for differences in the composition of the microbiomes among the experimental groups (see Stappenbeck & Virgin "Accounting for reciprocal host-microbiome interactions in experimental science" *Nature* 2016, 534: 191).

→ In the previous study (*J Allergy Clin Immunol* 2016; 137:507-16; Fig E1), we confirmed that *B. longum* administered orally was in its intact form in feces via FISH analysis. In addition, as per the reviewer's comment, when the animals were littermate-controlled, the effect of administered bacteria was seen more clearly. However, if bacteria are not effective in mice that are not littermate-controlled, we think that it will be less effective for humans with significantly different communities of gut microbiota between individuals, causing difficulties for combination therapy with IgE_{TRAP}.

J Allergy Clin Immunol 2016; 137:507-16; Fig E1

Reviewer #2 (Remarks to the Author):

An and colleagues present an antibody fusion molecule which combines the alpha chain of the high affinity IgE Fc receptor with an Fc domain comprising an IgD and IgG4 fusion. This molecule, alone or in combination with the probiotic bacterium *Bifidobacterium longum* was tested as a treatment for allergy in vitro and in vivo. This work is interesting, there are novel aspects to the study and the data show therapeutic promise in the systems tested. However, there are elements to the presentation of the study which do not allow for sufficient understanding or appraisal of the data and these need to be addressed.

- The choice of IgD domain in the Fc part of the construct and the combination with IgG4 domains is unclear and not justified anywhere in the manuscript. Why not use a full IgG4 Fc instead?

→ In accordance with the reviewer's comment, we have added the description for IgD/IgG4 hybrid Fc in the **revised Lines 104-110**.

- The schematic in Figure 1 is not sufficient to convince the reader that the Fc construct is designed to minimise the binding to Fc gamma Receptors. IgG4 Fc domains bind very well to Fc gamma Receptors and unless specific mutations are introduced the domain will still bind to Fc gamma Receptors including the high affinity Fc gamma RI. Are there mutations introduced in the construct, where and why? These should be explained in the manuscript and represented in the schematic in Figure 1A and 1B. A clear explanation should be included as to how the construct is designed to avoid binding to Fc gamma Receptors.

→ As per the reviewer's comment, IgG4 strongly binds FcγRs. However, because binding sites for FcγRs and C1q are located across the hinge proximal region and upper region of the CH2 domain, and IgD does not have these binding sites (unlike IgG, IgD/IgG4, which has a hinge proximal region and upper region of CH2 domain of IgD), it cannot bind to FcγRs and C1q. Moreover, results in **the revised Figure 1 D & E** show that IgD/IgG4 hybrid Fc does not bind to FcγRs and C1q without mutations. Further discussion has been added in **revised Lines 104-110**. In the revised figures, the following results (C1q binding data) were newly added.

- In the mast cell degranulation model:

o LAD-2 cells are poor degranulators, to convince the reader that these cells are triggered by IgE immune complexes in the first place, the % degranulation of these cells should be shown as controls in the in vitro model and the % degranulation should be compared with the blocking by omalizumab and by IgE(TRAP) at different concentrations of each drug.

o The data should be reproduced in rat basophilic leukemia cells.

→ As per the reviewer's comment, we have changed the graph of degranulation as follows and added it as **revised Figure 2D**.

→ Rat basophilic leukemia (RBL)-2H3 cells express rat FcεRα, which does not bind to human IgE. Moreover, because omalizumab does not block rat IgE, comparative experiments with omalizumab cannot be performed using RBL-2H3 cells. For this reason, we used human mast cell line LAD cells.

Table 1 | **Comparison of human and murine FcεRI**

Property	Murine FcεRI*	Human FcεRI*
Structure	Tetrameric αβγ ₂ structure only	Tetrameric αβγ ₂ structure and trimeric αγ ₂ structure
Expression pattern	Mast cells and basophils	Mast cells, basophils, monocytes and macrophages, myeloid dendritic cells, plasmacytoid dendritic cells, Langerhans cells, eosinophils and platelets
Regulation of expression	IL-4 does not enhance α-chain expression	IL-4 enhances α-chain expression
Binding properties	Binds murine IgE	Binds human and murine IgE

*REFS 1,9–11,13–17. FcεRI, high-affinity Fc receptor for IgE; IL-4, interleukin-4.

Nat Rev Immunol 2007;7:365-78

- In the introduction it is incorrectly stated that Omalizumab is the only anti-IgE agent approved for clinical use. This is incorrect. The authors should acknowledge that Ligelizumab has also been approved and this antibody has even higher affinity for the Cepsilon3 domain of IgE. This is also a humanized monoclonal IgG1k and would be expected to suffer from the same issues as Omalizumab. This should be acknowledged, discussed, and explained in the manuscript. No comparisons are made here between IgE(TRAP) and this very high affinity antibody, an agent like IgE(TRAP) is likely to be compared with Ligelizumab in future because of the high affinity to IgE

→ Ligelizumab is not yet approved by FDA as an allergy drug, but recently received FDA Breakthrough Therapy designation for CSU patients (**January 14, 2021**) and is expected to obtain FDA approval soon. Therefore, we have changed the words 'Only one IgE inhibitor' and 'the only a currently-available treatment' to 'an IgE inhibitor' and 'a currently-available treatment' in the explanation for omalizumab and for the most recent information added 'and recently received FDA Breakthrough Therapy designation for CSU patient' in the explanation for ligelizumab (**the**

revised Lines 252-254). As per the reviewer's comment, ligelizumab was already described in one paragraph of the discussion section (**the revised Lines 252-264 and 291-292**). However, unfortunately, we were not able to obtain ligelizumab and did not perform comparative experiments with IgE_{TRAP}.

Our Company □ Our Focus □ Our Impact □ Our Science □ Careers □

Novartis ligelizumab (QGE031) receives FDA Breakthrough Therapy designation for patients with chronic spontaneous urticaria (CSU)

□ Back to News Archive

Jan 14, 2021

- *Ligelizumab is the first treatment to receive FDA Breakthrough Therapy designation in chronic spontaneous urticaria (CSU) in patients with an inadequate response to H1-antihistamines¹*
- *Currently there are limited approved therapies for patients with CSU, also known as chronic idiopathic urticaria (CIU)*
- *Breakthrough Therapy designation suggests ligelizumab has the potential to provide a substantial benefit over existing available treatments*
- *U.S. regulatory filing in CSU is anticipated in 2022*

Basel, January 14, 2021 — Novartis today announced that the U.S. Food and Drug Administration (FDA) has granted ligelizumab (QGE031) Breakthrough Therapy designation for the treatment of chronic spontaneous urticaria (CSU), also known as chronic idiopathic urticaria (CIU), in patients who have an inadequate response to H1-antihistamine treatment.

- The authors claim that IgE(TRAP) did not elicit a positive helper T cell response in CD8+ EpiScreen assays using depleted peripheral blood mononuclear cells (PBMCs) from 50 healthy human donors, indicating that IgE_{TRAP} exhibits minimal immunogenicity. Given that lack of immune activation is a big claim in this study, the experimental design should be included in the methods section and the data should be shown.

→ According to the reviewer's comment, we have shown the results of the EpiScreen assay in **revised Table 1** and added the protocol in the methods section (**the revised Lines 351-360**).

Table 1 | EpiScreen assay

Sample	Mean SI*	SD	% Response
IgE _{TRAP}	N/A	N/A	0
PBS	N/A	N/A	0
Trastuzumab (Herceptin)	2.35	± 0.43	4
Exenatide (Bydureon)	2.47	± 0.64	14
Keyhole limpet hemocyanin (KLH)	9.15	± 8.75	90

*Stimulation Index (SI) = mean value of test wells (cpm) / baseline (cpm). The mean SI was calculated from the average of all positive donor responses observed during the entire time course (days 5-8). N/A indicates no data available.

- The data presented in cynomolgus monkeys are not justified sufficiently. Monkeys have normal levels of IgE in the blood and are not an allergy model. The authors claim that this drug could work in instances of high serum IgE levels in allergic patients being the competitive advantage of this new construct. Therefore, this monkey model will not be representative of a patient situation.

→ We agree with the reviewer's comment. Therefore, in order to avoid confusion, we have clearly described the reason for using non-allergic monkeys in the **revised Lines 143-145**.

An assay should be introduced to show lack of Fc-mediated functions of the construct, e.g. CDC or ADCC.

→ For ADCC or CDC experiments, a cell line highly expressing the antigen that IgE_{TRAP} recognizes, the IgE Fc domain, is needed as shown in the picture below. Unfortunately, this cell line is not currently available and we could not perform these experiments.

- The data showing differences in half-life of the constructs with high and low sialylation should be shown and explained in the discussion. Do the authors expect one of these variants to be a better therapeutic agent and why?

→ In accordance with the reviewer's comment, we have shown the results of pharmacokinetics experiments according to sialic acid content in **the revised Figure 3F and Table 2** and explained further in the results, discussion and methods sections (**the revised Line 156-170, 232-234, 305-307 & 398-401**).

Table 2 | Pharmacokinetics profiles according to sialic acid content of IgE_{TRAP}

Sialic acid content (mol/mol)	T _{1/2} (hr)	T _{max} (hr)	C _{max} (µg/mL)	AUC _{last} (hr*µg/mL)	AUC _{0-∞} (hr*µg/mL)
10.3	37.6	3	1.5 ± 0.5	62.0 ± 5.5	62.7
14.9	34.8	10	2.6 ± 0.2	197.3 ± 5.6	199.0
21.4	39.9	10	12.1 ± 0.7	953.9 ± 45.1	969.5

*T_{1/2}, half-life; T_{max}, Time of maximum concentration observed; C_{max}, maximum concentration observed; AUC_{last}, area under the curve from time zero to time of last measurable concentration; AUC_{0-∞}, area under the curve from time zero to infinity.

- What is the effect of the drug on human blood basophils?

→ Because basophils, like mast cells, are key effector cells for allergic responses and highly express FcεRα, the effect of IgE_{TRAP} on human blood basophils is the same as for mast cells. In order to avoid confusion, we have added an explanation for basophils in **the revised Lines 209-210**.

- The Abstract should include a justification for combination treatment with the probiotic bacterium. As written this combination is not clear to the general readership.

→ According per the reviewer's comment, we have justified the combination therapy approach in the abstract section (**the revised Lines 52-55**).

REVIEWER COMMENTS

Reviewer #1 (Remarks to the Author):

It looks like this revision was prepared in a hurry because the revised lines cited in the rebuttal letter do not correspond to the yellow highlighted lines in the revised manuscript. In addition many of the concerns raised by this reviewer have either not been addressed or have been dismissed.

1. I'm confused about the authors' response regarding Table 1. The data shown in the rebuttal letter (A) T cell proliferation and (B) No. IL-2 secreting cells is clearly taken from PLoS One 2016;11:e0159328.

Is Table 1 completely new to this manuscript, i.e. have the authors re-done the assay with these new controls? I'm not clear on why the authors have included the data from the PLoS One paper in the rebuttal.

2. The explanation for using non-allergic monkeys is not in revised lines 143-145. Please answer this question.

3. The response to the question regarding the viability of the *B. longum* administered is unacceptable.

In my original review I noted that the authors treated some mice with IgETRAP plus freeze dried *B. longum*. I asked if *B. longum* colonized the mice. The authors respond that in a previous publication they showed that *B. longum* could be detected in intact form in the feces via FISH analysis. This response does not answer any of my questions about the current manuscript. I asked specifically whether *B. longum* colonized the mice in the current study and whether the authors were able to recover live *B. longum* in the feces of the treated mice. The viability and colonization of *B. longum* needs to be validated in every single experiment that makes use of *B. longum* as a biotherapeutic. Historical controls are inadequate (and FISH analysis doesn't evaluate viability).

I don't understand the authors' response to my question about their failure to use littermate controls. I think they are saying that they want to demonstrate the bacteria are effective in mice that are not littermate controlled because humans have highly variable gut microbiomes. This is a valid concern for translating a potential biotherapeutic. However, before considering translation the authors need to clearly document efficacy in rigorously performed experiments in littermate-controlled mice.

No information at all is provided about *B. longum* in the Methods. What is the source? How was it grown? How did the authors determine CFUs? For the food allergy model the Methods states that freeze-dried *B. longum* was mixed with powdered mouse chow at 3×10^9 CFU/gram. Gram of what? Mouse chow? Were the mice fed ad libitum meaning that each mouse would receive a different dose? Later in the Methods the authors say that *B. longum* was administered at 5×10^9 cfu/head). What does this mean?

Reviewer #2 (Remarks to the Author):

The authors have addressed my comments.

Nature Communications Manuscript Number: NCOMMS-20-28339A

Reponses to the reviewer's comments for the manuscript entitled: "*Combination therapy with IgE_{TRAP} and probiotics effectively reduces IgE-mediated responses in an allergy model*"

REVIEWER COMMENTS

Reviewer #1 (Remarks to the Author):

It looks like this revision was prepared in a hurry because the revised lines cited in the rebuttal letter do not correspond to the yellow highlighted lines in the revised manuscript. In addition, many of the concerns raised by this reviewer have either not been addressed or have been dismissed.

1. I'm confused about the authors' response regarding Table 1. The data shown in the rebuttal letter (A) T cell proliferation and (B) No. IL-2 secreting cells is clearly taken from PLoS One 2016;11:e0159328.

Is Table 1 completely new to this manuscript, i.e. have the authors re-done the assay with these new controls? I'm not clear on why the authors have included the data from the PloS One paper in the rebuttal.

→ We are sorry to confuse you. Table1 shows completely new results from the experiment re-done using trastuzumab (Herceptin), exenatide (Bydureon) and keyhole limpet hemocyanin (KLH) as controls.

2. The explanation for using non-allergic monkeys in not in revised lines 143-145. Please answer this question.

→ Because omalizumab does not recognize mouse IgE, it is difficult to compare the ability of IgE_{TRAP} and omalizumab to block IgE in a mouse model. Moreover, inducing allergy in monkeys is difficult to do with lack of a well-characterized model that is widely used and repeated by peers. Therefore, we focused on the abilities of IgE_{TRAP} and omalizumab to control high IgE levels in serum by using monkeys with high serum IgE levels under basal condition. For the better understanding of readers, additional explanations were included in the **revised line 133-136 (Results section)**.

3. The response to the question regarding the viability of the *B. longum* administered is unacceptable.

In my original review I noted that the authors treated some mice with IgETRAP plus freeze dried *B. longum*. I asked if *B. longum* colonized the mice. The authors respond that in a previous publication

they showed that *B. longum* could be detected in intact form in the feces via FISH analysis. This response does not answer any of my questions about the current manuscript. I asked specifically whether *B. longum* colonized the mice in the current study and whether the authors were able to recover live *B. longum* in the feces of the treated mice. The viability and colonization of *B. longum* needs to be validated in every single experiment that makes use of *B. longum* as a biotherapeutic. Historical controls are inadequate (and FISH analysis doesn't evaluate viability).

→ According to the reviewer's comment, we examined whether *B. longum* colonize mice by culturing of gut microbiota and subsequent genomic analysis. We analyzed the fecal and intestinal samples of mice administered with *B. longum* versus control and sequenced the fecal and intestinal contents. We also cultured these samples and sequenced the microbes. These data are shown in the **revised Supplemental Figure 4 & 5** and the **revised Supplementary Table 1**, and explained in the **revised Line 184-198 (Results section)** and the **revised Line 281-288 (Discussion section)**.

Supplementary Figure 4.

A

B

C

Supplementary Figure 5.

Supplementary Table 1. Ratio of reads assigned to Bifidobacterium

Sample	Fecal			Cecum (contents)			Cecum (tissue)		
	Control (PBS)	B. longum Low (1x10 ⁹ cfu)	B. longum High (1x10 ¹⁰ cfu)	Control (PBS)	B. longum Low (1x10 ⁹ cfu)	B. longum High (1x10 ¹⁰ cfu)	Control (PBS)	B. longum Low (1x10 ⁹ cfu)	B. longum High (1x10 ¹⁰ cfu)
Littermate #1	0	0	0.0013 %	0	0	0	0	0	0
Littermate #2	0	0	0	0	0	0.0012 %	0	0	0.0091 %
Littermate #3	0	0	0	0	0	0	0	0	0.0036 %
Littermate #4	0	0.0011 %	0	0	0	0	0	0	0
Littermate #5	0	0	0	0	0	0	0	0	0
Littermate #6	0	0	0	0	0	0	0	0	0

Supplementary Table 1.

I don't understand the authors' response to my question about their failure to use littermate controls. I think they are saying that they want to demonstrate the bacteria are effective in mice that are not littermate controlled because humans have highly variable gut microbiomes. This is a valid concern for translating a potential biotherapeutic. However, before considering translation the authors need to clearly document efficacy in rigorously performed experiments in littermate-controlled mice.

→ We acknowledge that the reviewer made a valid point here. Therefore, in the experiments that we newly performed to sequence *B. longum*'s presence in the mice given probiotics, we used litter-matched mice for test and control groups.

No information at all is provided about *B. longum* in the Methods. What is the source? How was it grown? How did the authors determine CFUs?

→ We had described information regarding the source of *B. longum* and measurement of CFUs in the **revised Line 400-403 (Materials and Methods section)**

For the food allergy model the methods states that freeze-dried *B. longum* was mixed with powdered mouse chow at 3x10⁹ CFU/gram. Gram of what? Mouse chow?

→ It is 3x10⁹ CFU/gram of powdered mouse chow.

Were the mice fed ad libitum meaning that each mouse would receive a different dose? Later in the methods the authors say that *B. longum* was administered at 5×10^9 cfu/head). What does this mean?

→ In the case of long-term feeding experiments using Zonde (=feeding needle), stress may affect the experimental results. For this reason, we initially chose ad-libitum feeding over the forced feeding in order to prevent the stress of the mice affecting the results. Unfortunately, the observed variation in the food intake between the experimental mice were greater than we had expected. We chose the intragastrical feeding in the later experiments. Therefore, to avoid confusion between these two feeding methods, feeding method was accurately described in **the revised Line 395-400 and 409-410 (Material and Method section)**.

Reviewer #2 (Remarks to the Author):

The authors have addressed my comments.

An assay should be introduced to show lack of Fc-mediated functions of the construct, e.g. CDC or ADCC.

→ According to the reviewer's comment, we performed the experiment related to ADCC. The data is now shown as the **revised Figure 1D**, and explained in the **revised Line 103-106 (results section)** and in **the revised Line 323-326 and 356-366 (Materials and Methods section)**.

Figure 1

REVIEWER COMMENTS (2nd ROUND REVISION)

Reviewer #1 (Remarks to the Author):

It looks like this revision was prepared in a hurry because the revised lines cited in the rebuttal letter do not correspond to the yellow highlighted lines in the revised manuscript. In addition, many of the concerns raised by this reviewer have either not been addressed or have been dismissed.

1. I'm confused about the authors' response regarding Table 1. The data shown in the rebuttal letter (A) T cell proliferation and (B) No. IL-2 secreting cells is clearly taken from PLoS One 2016;11:e0159328.

Is Table 1 completely new to this manuscript, i.e. have the authors re-done the assay with these new controls? I'm not clear on why the authors have included the data from the PLoS One paper in the rebuttal.

→ We apologize for the confusion. The data from the EpiScreen assay in **the revised Table 1** was newly obtained by our group.

2. The explanation for using non-allergic monkeys is not in revised lines 143-145. Please answer this question.

→ This comment is the same as in 1st round revision. So, please see **the answer to comments #2** in 1st round revision.

3. The response to the question regarding the viability of the B. longum administered is unacceptable.

In my original review I noted that the authors treated some mice with IgETRAP plus freeze dried B. longum. I asked if B. longum colonized the mice. The authors respond that in a previous publication they showed that B. longum could be detected in intact form in the feces via FISH analysis. This response does not answer any of my questions about the current manuscript. I asked specifically whether B. longum colonized the mice in the current study and whether the authors were able to recover live B. longum in the feces of the treated mice. The viability and colonization of B. longum needs to be validated in every single experiment that makes use of B. longum as a biotherapeutic. Historical controls are inadequate (and FISH analysis doesn't evaluate viability).

I don't understand the authors' response to my question about their failure to use littermate

controls. I think they are saying that they want to demonstrate the bacteria are effective in mice that are not littermate controlled because humans have highly variable gut microbiomes. This is a valid concern for translating a potential biotherapeutic. However, before considering translation the authors need to clearly document efficacy in rigorously performed experiments in littermate-controlled mice.

→ We acknowledge that the reviewer made a valid point here. Since this comment is the same as in 1st round revision, please see **the answer to comments #6** in 1st round revision.

4. No information at all is provided about *B. longum* in the Methods. What is the source? How was it grown? How did the authors determine CFUs?

→ We added the information regarding the *B. longum* in **lines 435-442 of the revised Materials and Methods section and Supplementary Methods of the revised Supplementary Information.**

5. For the food allergy model the methods states that freeze-dried *B. longum* was mixed with powdered mouse chow at 3×10^9 CFU/gram. Gram of what? Mouse chow?

→ The freeze-dried *B. longum* was given by mixing 3×10^9 CFU/gram of powdered mouse chow. This is described in **lines 451-458 of the revised Materials and Methods section.**

6. Were the mice fed ad libitum meaning that each mouse would receive a different dose? Later in the methods the authors say that *B. longum* was administered at 5×10^9 cfu/head). What does this mean?

→ Although we mixed the food and *B. longum* appropriately considering the average daily food intake of mice and the sensitivity of *B. longum* contact, as the reviewer said, mice are very likely to eat *B. longum* in different doses. So, for the subsequent experiments after the initial experiment, intragastrical feeding was used to administer the correct amount of *B. longum*. This is described in **lines 451-458 of the revised Materials and Methods section.**

Reviewer #2 (Remarks to the Author):

The authors have addressed my comments. → There were no additional comments.

REVIEWER COMMENTS

Reviewer #1 (Remarks to the Author):

The authors have made a valiant effort to address the concerns raised in the initial review. Most of the new data presented is compelling but at some points they have sidestepped the concerns that have been raised.

First the authors propose that IgETRAP will be both safer and more effective than omalizumab.

I asked them to test IgETRAP and omalizumab side by side in their EpiScreen assay. Instead, they repeated the assay using trastuzumab (which is NOT a monoclonal antibody that binds to IgE) instead of omalizumab because “trastuzumab appeared to have a slightly weaker response than omalizumab according to a previous report.” I don’t understand the meaning of this response and maintain that the direct comparison of omalizumab and IgETRAP in the EpiScreen assay is necessary for this report.

Regarding the authors comments on free IgE, total IgE and antigen specific IgE this reviewer is not confused by these terms. Indeed, I think it is very important to show that IgETRAP can raise the concentration of OVA-specific IgE (as shown in the “reviewer only” figure). These reviewer only figures and the authors’ interpretation of this data should be added to the manuscript. Data on peanut specific IgE should be added to the new Figure 6.

In response to my questions about detection of *B. longum* the authors show that *B. longum* does NOT colonize and should state this clearly. They can culture *B. longum* from feces 4 hrs after oral administration so they have shown that they have administered viable bacteria. They say that their results show that “*B. longum* enhances the therapeutic effect of IgETRAP by acting directly rather than altering the intestinal microbial community.”

It is not clear to this reviewer what this statement means.

Nature Communications Manuscript Number: NCOMMS-20-28339D

Reponses to the reviewer's comments for the manuscript entitled: "*Combination therapy with IgE_{TRAP} and probiotics effectively reduces IgE-mediated responses in an allergy model*"

REVIEWER COMMENTS

Reviewer #1 (Remarks to the Author):

The authors have made a valiant effort to address the concerns raised in the initial review. Most of the new data presented is compelling but at some points they have sidestepped the concerns that have been raised.

1. First the authors propose that IgETRAP will be both safer and more effective than omalizumab. I asked them to test IgETRAP and omalizumab side by side in their EpiScreen assay. Instead, they repeated the assay using trastuzumab (which is NOT a monoclonal antibody that binds to IgE) instead of omalizumab because "trastuzumab appeared to have a slightly weaker response than omalizumab according to a previous report." I don't understand the meaning of this response and maintain that the direct comparison of omalizumab and IgETRAP in the EpiScreen assay is necessary for this report.

→ In accordance with the reviewer's comment, we performed an additional EpiScreen assay that directly compares IgE_{TRAP} with omalizumab, and the results of the independently conducted two experiments are shown in Table 1, Supplementary Table 1, and Supplementary Fig.1. The related descriptions are in **lines 107-114 of the revised Results section, lines 401-420 of the revised Materials and Methods section and lines 265-271 of the revised Discussion section.**

2. Regarding the authors comments on free IgE, total IgE and antigen specific IgE this reviewer is not confused by these terms. Indeed, I think it is very important to show that IgETRAP can raise the concentration of OVA-specific IgE (as shown in the "reviewer only" figure). These reviewer only figures and the authors' interpretation of this data should be added to the manuscript. Data on peanut specific IgE should be added to the new Figure 6.

→ According to the reviewer's comment, we added the experimental results showing the antigen-specific IgE (OVA-specific IgE and peanut-specific IgE) as well as total IgE to Fig. 6d and Supplementary Fig. 4. The related descriptions are in **lines 183-187 and 196-197 of the revised Results section, lines 441-449 of the revised Materials and Methods section and lines 260-264 of the revised Discussion section.**

3. In response to my questions about detection of *B. longum* the authors show that *B. longum* does NOT

colonize and should state this clearly. They can culture *B. longum* from feces 4 hrs after oral administration so they have shown that they have administered viable bacteria. They say that their results show that “*B. longum* enhances the therapeutic effect of IgETRAP by acting directly rather than altering the intestinal microbial community.”

It is not clear to this reviewer what this statement means.

→ As the reviewer said, we clearly stated in **the lines 220-222 of the revised Results section and the lines 313-315 of the revised Discussion section** that *B. longum* did not easily colonize the host gut. In addition, we further stated in **the lines 211-212 of the revised Results section** that result of fecal culture showed that *B. longum* was administered in a live state. Description of the action of *B. longum* itself has been added to **the lines 315-321 of the revised Discussion section**.

REVIEWERS' COMMENTS

Reviewer #1 (Remarks to the Author):

Thank you for your thorough response. All of my concerns have been addressed.